

# From wind conditions to operational strategy: Optimal planning of wind turbine damage progression over its lifetime

Niklas Requate[1], Tobias Meyer[1], and René Hofmann[2]

[1]Fraunhofer IWES, Bremerhaven, Germany
[2]TU Wien, Vienna, Austria

**Correspondence:** Niklas Requate (niklas.requate@iwes.fraunhofer.de)

**Abstract.** Renewable energies have an entirely different cost structure than fossil fuel-based electricity generation. This is mainly due to the operation at zero marginal cost, whereas for fossil fuel plants, the fuel itself is a major driver of the entire cost of energy. For a wind turbine, most of the materials and resources are spent up front. Over its lifetime, this initial capital and material investment is converted into usable energy. Therefore, it is desirable to gain the maximum benefit from the utilized 5 materials for each individual turbine over its entire operating lifetime. Material usage is closely linked to individual damage progression of various turbine components and their respective failure modes.

Within this work, we present a novel approach for an optimal long-term planning of the operation of wind energy systems over their entire lifetime. It is based on a process for setting up a mathematical optimization problem that optimally distributes the available damage budget of a given failure mode over the entire lifetime. The complete process ranges from an adaptation 10 of real-time wind turbine control to the evaluation of long-term goals and requirements. During this process, relevant deterministic external conditions and real-time controller setpoints influence the damage progression with equal importance. Finally, the selection of optimal planning strategies is based on an economic evaluation. The method is applied to an example for demonstration. It shows the high potential of the approach for an effective damage reduction on different use cases. The focus of the example is to effectively reduce power of a turbine under conditions where high loads are induced from wake-induced 15 turbulence of neighbouring turbines. Through the optimization approach, the damage budget can be saved or spent under conditions where it pays off most in the long-term perspective. This way, it is possible to gain more energy from a given system and thus to reduce cost and ecological impact by a better usage of materials.

## 1 Introduction

Meeting the rising demand for energy without using fossil fuels is probably one of the greatest challenges of our time. Wind 20 energy plays a key role in achieving this worldwide, and the wind industry has been developing to a mature and effective branch of technology. Nevertheless, energy production will always involve the use of materials and resources. For a wind turbine, this includes the production of large complex components like the tower, the rotor blades and the generator, but also the use of land on- and offshore as well as continuous operating costs due to maintenance and repair activities.



Therefore, it is desired to gain the maximum benefit from the utilized materials for each individual turbine over its entire operating lifetime. The materials will be used up through the operation in many different ways. The usage is closely linked to individual failure modes of various turbine components. While some of these failure modes need to be avoided through advancements in design and robustness to environmental conditions, other failure modes are highly influenced through the operational strategy. Especially fatigue damage is strongly influenced by induced loads which depend on the external conditions in combination with the operational control of the turbine. Even with the smartest individual control solutions for load reduction like e.g. individual pitch control and active damping, there will always be some trade-off between power production and induced damage which cannot be fully prevented. Additionally, load reducing effects for some failure modes might have negative effects on others.

With the experience and development of a maturing wind industry, standard procedures for the design of wind turbines have been established for finding a reasonable trade-off between induced damage and power production. This way, wind turbines be operated for at least 20 years under various conditions from the environment and the grid. While the external conditions of each turbine are highly individual, wind turbine design can only consider site-specific conditions to some extent, e.g. by type certificates for different wind classes IEC (2019). In order to operate each turbine at its individual optimal balance of induced damage and power production, an adaptive operation based on information of the current condition and performance is required. A concept for such an operation is proposed through reliability(-adaptive) control which can principally be applied to any system where components are used up from operation, i.e. are subject to degradation. The reliability controller is implemented as a closed-loop supervisory controller which adapts the system such that it meets predetermined reliability objectives. Within this concept, it is important to distinguish between the real-time controller directly interacting with the actuators of a system and the outer supervisory control. The outer loop runs on a slower time scale and can send setpoints to the real-time controller.

Within this work, a method for finding an optimal long-term operational planning which already includes the available setpoints for the wind turbine real-time controller is presented. Thus, it contributes to the development of a reliability(-adaptive) control loop for wind turbines by creating a desired operation which is necessary for a closed-loop operation. It also brings advantages in itself for an open-loop operation. Within this chapter, the method is first embedded into the context of reliability(-adaptive) control (Sect. 1.1). Afterwards, the objectives of this work are further elaborated in Sect. 1.2 and the methodology for creating an optimal planning is explained in Sect. 1.3.

## 1.1 State of the art

A concept for a Safety and Reliability Control Engeneering (SRCE) including a supervisory reliability controller, which uses information about the current state-of-health, was introduced in Söffker and Rakowsky (1997) and further discussed e.g. in Rakowsky (2005) and Rakowsky (2006). In Meyer (2016) a reliability controller based on the health index, used as a measure for the state-of-health, for a mechatronic system was implemented and validated. On the one hand, the application of such an approach for wind energy systems has a high potential due to the highly individual site and turbine specific operating and environmental conditions as well as ageing characteristics of various components (Meyer et al., 2017). On the other hand,



the complexity of the coupled system, the interaction of wind turbines in a wind farm as well as constraints from operating and maintenance strategies, market conditions, grid requirement and nevertheless certification processes lead to a challenging

interaction of different areas. One of the major aspects for the operation of a reliability controller in a closed-loop is the information about the state-of-health of the considered system. While wind turbines are equipped with various sensors and associated condition monitoring systems (CMS) or structural health monitoring (SHM) systems, the prognosis of the actual state-of-health and the associated remaining useful lifetime (RUL) still requires a lot of research and development. In Beganovic and Söffker (2016), an overview of signal-based monitoring methods with a focus on the usage for online fault detection and

advanced control is provided. In Do and Söffker (2021), an overview of management and control strategies for wind turbines based on health prognostics is provided. Both papers clearly state that further investigation is needed to determine the state-of-health. Additionally, the high requirements for an adaptive controller due to the multi-objective nature of the problem under various loading condition is also mentioned. Nevertheless, the full advantage of health monitoring combined with advanced reliability control strategies can only be fully exploited with further development in each of the fields, which can later be

combined to an integrated approach.

There are two major advantages which result from the use of closed-loop structure for controlling the reliability. On the one hand, it enables a synchronization with maintenance strategies or planned decommissioning. On the other hand, it allows extending the lifetime of a system by switching to a load reducing control configuration at any point in time. The latter point is specifically addressed in the concept of life extending control, where a concept was introduced in Lorenzo and Merrill

(1991). This concept is more oriented towards fatigue damage and thus also well applicable for wind turbines. The approach was pursued for wind turbine operation in Santos (2006) and the associated patent (Santos, 2008). In the study, the wind turbine actuators are directly modified by a model predictive control algorithm, which receives setpoints for the degradation of the turbine from a supervisory control loop. Comparable concepts based on an online fatigue accumulation using online rainflow counting were also followed by Loew et al. (2020) and Njiri et al. (2019). The latter is clearly related to the concept of

reliability adaptive control, which was explained above. In all three of the applications, the controllers are tested on rather short timeframes of at maximum 600 seconds so that long-term benefits from the methods can not yet be fully considered. Long-term effects of adapting control strategies during operation for lifetime extension are examined in Pettas and Cheng (2018) and Pettas et al. (2018). In Requate and Meyer (2020), the concept of reliability control is implemented by switching between different down- and uprating configurations to follow a predetermined desired degradation for several years. Dependent on the

desired target, a lifetime extension by several years can be reached. While the concept of directly adapting the turbine actuators according to the desired planning targets might have a higher theoretical potential because its reaction is more flexible, the concept of switching between different configurations seems to be more straightforward to implement for existing structures for wind turbine and wind farm control concepts. It also facilities a guarantee for a safeguarded operation in all of the selected configurations. The combination of both concepts might offer additional advantages in the future.

In all of the mentioned work, the aspect of planning the operation up until the end of a wind turbine's lifetime has not yet been addressed in much detail. This becomes even more relevant in the context of wind farm control where the higher level constraints like the market prices, maintenance strategies, planning of decommissioning are relevant. Wind farm control has





gained growing interest of research and also industry in recent years. One major focus of research was the mitigation of wake effects, which decrease power production but increase loads on downstream turbines (Dimitrov, 2019). Wake steering by yaw,

but also derating[1] of the upstream turbines can be used, to increase the overall power production of a wind farm. In addition to increasing the power production, the influence on the loads and lifetime of the wind turbines of such methods are also examined (Andersson et al., 2021; Nash et al., 2021; Meyers et al., 2022; Houck, 2022). At first, the focus is not to increase the loads above the limits of certification, but the use of wind farm control for active load reduction is also examined in several studies (Bossanyi, 2018; Kanev et al., 2018, 2020; Harrison et al., 2020). Concepts for an integrated control of wind farms

covering the complete range from short-term demands for grid services up to long-term objectives for reliability are required (Eguinoa et al., 2021; Kölle et al., 2022). Therefore, combining the approaches of wind farm control with reliability adaptive control offers a high potential for a truly optimal operation, e.g. by intelligently managing which turbines should take over grid services in certain situations based on their current state-of-health and a planning until the end of the desired lifetime. For future energy systems, the interconnection to storage systems or power-to-X technologies and their reliability and degradation

mechanisms also need to be considered.

Since the future damage progression of a system depends on the way it is operated, it is important to integrate the adaptive control behaviour into the planning process. Implicitly, this is done when sector management is applied to avoid high loads from an upstream turbine. Previous studies have shown, that it is possible to balance energy and loads with sector management strategies using derating (Bossanyi and Jorge, 2016).

A method for derating a wind turbine is integrated into any modern wind turbine to comply with grid requirements in one way or the other. Additionally, it can be used as an instrument to either reduce the effects from wake on the downstream turbine or to reduce loads of the turbine itself. The derating of the turbine is a setpoint to the wind turbine's real-time controller. The implementation of the derating method by parameters within the real-time controller thus depends on the objective and also on the individual dynamic behaviour of each turbine (Meyers et al., 2022; Houck, 2022). Even reducing damage from heavy rain

on the leading edge of the blades might be a possible objective for rotor speed reduction, besides the more common fatigue damage (Bech et al., 2018). Other studies that are more focused on the specific objectives of this paper are presented at the appropriate part in the text, mainly in Sect. 2. The objectives are defined in the next section.

### 1.2 Objectives

Overall, we have identified three major requirements to allow for a reliability-adaptive closed-loop operation. At first, methods

for load reduction which reduce performance or conflict with other objectives need to be available within the wind turbine real-time controller. This requirement is definitely fulfilled for real-time controller wind turbines and wind energy systems, e.g. by using derating. The main difficulty lies in determining the amount of damage reduction for the individual failure modes of a turbine under various external input conditions. This is directly connected to the second requirement: The closed-loop

---

[1]In general, there are various terms for reducing the power of a wind turbine and the usage often depends on the context. In wind farm control, the term axial-induction control is often used. Also down-regulation or curtailment are prominent terms. The latter is often referred to in the context of requests by the grid operator. The term derating is used throughout this work.





controller needs continuous feed-back on the performance of the turbine, ideally on the state-of-health for each failure mode.
This involves further research and development on condition and health monitoring, which is an ongoing process as discussed
in Sect. 1.1. The third requirement consists of a planned desired operation of the system up until the desired lifetime, which is
addressed within this work.

We assume a basic setup for a supervisory reliability control loop of a wind turbine or a complete wind farm by separating
into different stages acting on different time scales. On the real-time stage, the dynamic loads of a wind turbine result from the
interaction between the real-time wind turbine controller and the external conditions from the environment and the grid. Those
loads slowly induce damage to the wind turbine. The supervisory reliability control loop acts on a time scale of 10 minutes up to
several days because such a time scale allows for an appropriate performance evaluation of the wind turbine in terms of damage
progression and power production. On this operating stage, setpoints are sent to the real-time controller of the wind turbine.
The planned desired operation determines the targets for this stage which result from the long-term reliability objectives, i.e.
the planned damage progression. Because of the dependency between reliability and operation, the desired operation already
needs to consider the influence of adaptive control on the damage while at the same time focusing on long-term planning
decisions and economic benefits. An overview of a wind farm which is operated using adaptive operation on these two stages
is given in Fig. 1. The long-term planning (Planning stage on left-hand side of the figure) can either be used in an open-loop by
providing setpoints to the wind turbine controller for specific input conditions or a target damage progression of the reliability
control loop. In both cases, it should cover most relevant deterministic effects on long-term damage progression in an optimal
way. Through a closed-loop behaviour on the operating stage, it is additionally possible to react to the actual performance of
the wind turbine, including the current state-of-health and additional current inputs from weather or market price conditions
(Right-hand side of the figure). At the same time, the long-term objectives are still met. A re-adaption of the planning required
when large deviations of the original plannings occur or if the long-term objectives change. Thus, it is not a real closed loop
operation, but it can also be applied when open-loop setpoints are sent to the real-time controller. It should just be applied after
longer time periods of months or several years.

The long-term objectives for wind turbine operation are specifically driven by fatigue damage progression, which is an
important failure mode for wind turbine principal components like the tower and the blades. For an optimal material usage,
fatigue budget is ideally fully used up at the desired lifetime while a maximum amount of energy has been produced during this
time. Thus, balancing the trade-off between induced damage and power production over the whole range of external conditions
and under consideration of their frequency of occurrence is required. The goal is to find a planning method, which distributes
the fatigue damage optimally over the planned operating time by saving the fatigue budget where it pays off most, i.e. where
loads are high, but energy production is low. This is possible because of the nonlinear relationship between external conditions,
load reducing control features and induced damage. When a turbine is subjected to high wake induced turbulence, for example,
the relationship between induced damage and produced energy is definitely worse than for a turbine operating at the same wind
speed at a low turbulence. The key question for an optimally planned target distribution is to decide by how much the damage
should be reduced through adaptive control so that the long-term objectives are met. To answer this question, a method to find
an optimal planning through mathematical optimization for an individual wind turbine is built up.





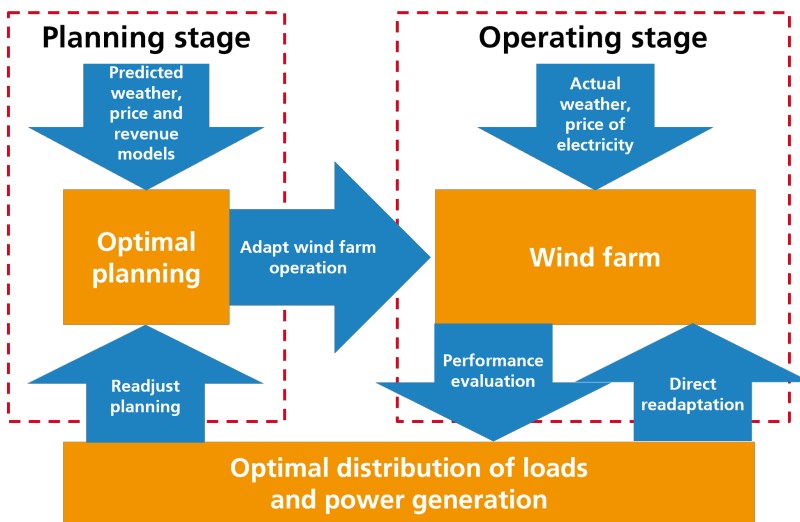

**Figure 1.** Overview of adaptive wind farm operation separated in to planning and operating stage

### 1.3 Methodology

In order to create a planning method which fulfills the objectives, the complete process from an adaptation of real-time wind turbine control to the evaluation of long-term goals and requirements needs to be covered. During this process, the influence on damage progression of relevant deterministic external conditions is just as important as that of real-time controller setpoints.

The key part of our proposed method consists of the formulation of a mathematical optimization problem, where the aim is to meet long-term objectives, such as maximum power or revenue over the entire lifetime, by finding an individual trade-off 165 between induced damage and power production for each relevant operational condition.

For application of our method to a given system, it is crucial to know how it interacts with its environment. For this, the system boundary must be well-defined beforehand. It forms the basis for definition of environmental inputs, for setpoints for the real-time controller, as well as for the damage of different failure modes and performance measures such as energy production.

We identified a four-step process to create the optimal planning *for this well-defined system* within its boundaries:

**Step 1: Provide adaptable real-time controller of the wind turbine**

The setpoints of the real-time controller of the system directly influence the trade-off between induced damage and energy production. To make use of this, several sets of setpoints need to be provided such that different trade-offs can be selected accordingly.

**Step 2: Build surrogate models for damage progression and energy production**

In order to optimally distribute the fatigue budget over the system's lifetime, it must be possible to evaluate the effect of changes in the optimization variables with little computational cost. This necessitates the use of surrogate models, which





represent the relationship between external conditions and setpoints of the controller to damage and energy. Thus once the real-time controller is finalized, such surrogate-models need to be setup.

**Step 3: Determine optimal condition-based operational strategies for lifetime planning**

By combining the surrogate models with a statistical frequency distribution of the relevant external conditions defined by the system boundaries, a rough long-term prediction of the damage progression can be obtained. This model of damage progression forms the basis of an optimization problem, in which the objective is to optimally distribute the available damage budget over the entire operating time. The optimization variables, i.e., the means to influence system operation, are the real-time controller setpoints for all external conditions. They are optimized for all external conditions at once.

This way, performance is maximized while the total damage budget is optimally distributed.

**Step 4: Select economically best operational lifetime-planning strategy**

     The optimization from step 3 yields multiple results, where each one represents an individual trade-off between energy production and damage. The selection of a single optimal planning becomes possible, by evaluating economic aspects of the results from step 3.

The four steps not only allow for a feasible computation time, but they also lead to an easily explainable result after each step, which is in stark contrast to more integrated approaches. The four steps can principally be applied to any system which is subject to a strong coupling of control setpoints and external conditions. Due to the high influence of wind conditions on the fatigue damage of wind turbines, wind energy systems represent a prime example for its application. Within this work, we concentrate on fatigue damage because it is the most significant failure mode for long-term operation of wind turbines (as 195 already mentioned in Sect. 1.2).

### 1.4    Outline of the remaining paper

The above mentioned four-step process form the core of the remaining paper. At first, their theoretical background and a more in-depth explaination are given in Sect. 2. Afterwards, the process is applied to an application example, a small wind farm with 9 generic 7.5 MW wind turbines. The focus of the example is to effectively reduce power of a turbine under conditions where 200 high loads are induced from wake-induced turbulence of a neighbouring turbine. The system is introduced and its boundaries are defined in Sect. 3. Afterwards, the steps are applied to the example in Sect. 4. The applied process and the results are discussed in Sect. 5 before the findings are concluded and an outlook is given in Sect. 6.

## 2    Theoretical background to application of four-step planning procedure

The basic idea of our method is to optimally distribute the induced damage over the operating time. With this, we assume a 205 continuous increase in damage over time, as depicted in Fig. 2. *Damage* always refers to damage which directly and exclusively contributes to a certain failure mode $fm$. The lifetime of a system or a component is reached when the damage for a failure mode $D_{fm}$ reaches the value 1, which is equivalent to 100% of the available damage budget. Using a reference operational strategy



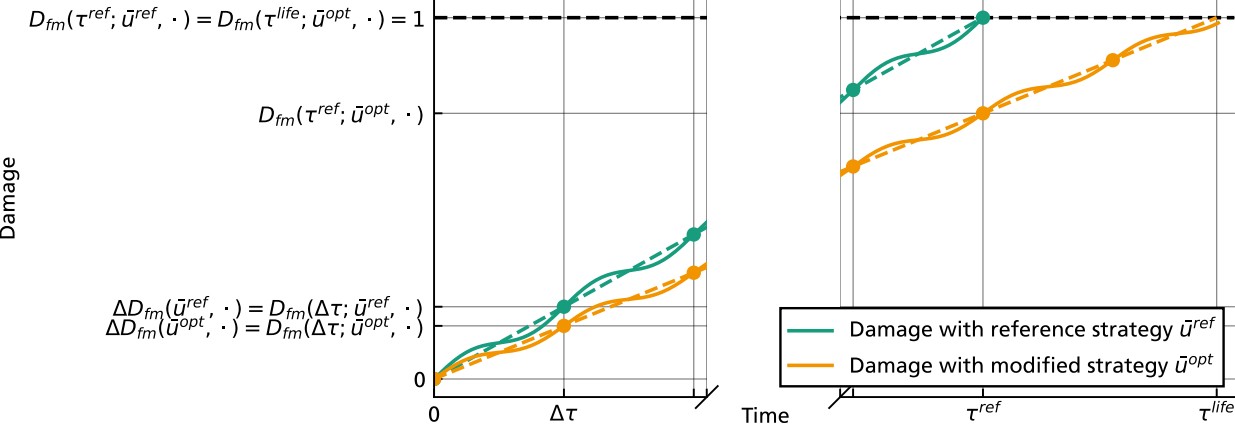

**Figure 2.** Illustration of damage progression over time for a reference and a modified operational strategy

$\bar{u}^{ref}$, the value[2] $D_{fm}(\tau^{ref};\bar{u}^{ref},\cdot)$ is reached at the reference lifetime $\tau^{ref}$. Our goal is to use a modified operational strategy $\bar{u}^{opt}$ over a freely chosen operation period $\tau^{life}$ to distribute the damage $D_{fm}(\tau^{life};\bar{u}^{opt},\cdot)$ in such a way that maximum

energy yield or largest economic profit is obtained. Fig. 2 also depicts a time increment $\Delta\tau$, which for wind energy problems is commonly selected as one calendar year. Within the time increment $\Delta\tau$, the damage rate is not constant. Instead, it changes over time due to e.g., seasonal variation of environmental conditions, and correspondingly varying setpoints. The continuous damage progression at the more detailed time scale is indicated in Fig. 2 by the wave-like behaviour of the increasing damage value. This relationship from environmental input conditions and setpoints to damage rate is highly nonlinear in both dimensions,

which makes it possible to compensate for high-damage environmental conditions by using low-damage setpoints. For now, the effect of seasonal variation on damage and energy yield is fully included in the final value after one time increment $\Delta\tau$. We use this as the basis of our optimization.

It is immediately apparent that for given values at discrete time increments $\Delta\tau$ only, a linear relationship between the damage $D_{fm}(\tau;\bar{u},\cdot)$ and time $\tau = Y \cdot \Delta\tau$ exists for any operational strategy $\bar{u}$. The value $Y$ is the number of time increments to the

full time period $\tau$, e.g., the number of operating years when $\Delta\tau$ represents one year. This is expressed by

$$D_{fm}(\tau;\bar{u},\cdot) = D_{fm}(\Delta\tau;\bar{u},\cdot) \cdot Y = \Delta D_{fm}(\bar{u},\cdot) \cdot \tau, \tag{1}$$

where $\Delta D_{fm}(\bar{u},\cdot)$ is the damage increment over one time increment $\Delta\tau$. We now assume that using an optimal operating strategy $\bar{u}_{opt}$, we achieve an optimized lifetime $\tau^{life}$. During this changed lifetime, the entire damage budget is spent, i.e., $D_{fm}(\tau^{life};\bar{u}^{opt},\cdot) = 1$. The modified lifetime period using $\bar{u}^{opt}$ is then simply given by inserting the optimized values in

---

[2]Note that in our notation which we distinguish between variables, i.e., values that can be changed, and parameters, which are fixed. They are separated by a semicolon. If additional variables or parameters exist but are not important for a certain passage, we omit them to improve readability and replace them with a central dot $\cdot$.





Eq. (1) and resolving for $\tau^{life}$:

$$\tau^{life} = \frac{D_{fm}(\tau^{life}; \bar{u}^{opt}, \cdot)}{\Delta D_{fm}(\bar{u}^{opt})} = \frac{1}{\Delta D_{fm}(\bar{u}^{opt}, \cdot)}. \tag{2}$$

Thus, our aim is now to find a strategy $\bar{u}^{opt}$ which optimally changes the damage increment to $\Delta D_{fm}(\bar{u}^{opt}, \cdot)$ within the fixed period $\Delta\tau$. The damage increment generally depends on environmental conditions, e.g., incoming wind conditions, and setpoints of any applied strategy $\bar{u}$.

Computing the modified lifetime with Eq. (2) can result in *any* timespan $\tau^{life}$. However, due to seasonality and the associated nonlinearity, which is depicted as solid curve in Fig. 2, *within* the time increment $\Delta\tau$, Eq. (1) only holds true for $Y \in \mathbb{N}$. This usually applies for $\tau^{ref}$, but the resulting value for $\tau^{life}$ from Eq. (2) depends on the optimized damage increment $\Delta D_{fm}(\bar{u}^{opt}, \cdot)$, which can take up any value, and is in turn not restricted to natural numbers. However, for a longer timespan $\tau^{life}$ the resulting error is small compared to the uncertainties resulting from the overall extrapolation process, and can be 235 neglected when the modified lifetime is computed.

That the assumption

$$\tau^{ref} = Y \cdot \Delta\tau, Y \in \mathbb{N}. \tag{3}$$

holds is due to a suitable scaling of $\Delta D_{fm}(\bar{u}^{ref}, \cdot)$. Among other things, this includes the assumptions that the damage budget is completely used up under the reference strategy $\bar{u}^{ref}$ and that the damage increment is always the same for each time increment 240 $\Delta\tau$. The latter is based on the standard approach in the design process of wind turbines, where the damage progress during one year is extrapolated to periods longer than 20 years. This is realized by using frequency distributions of the input conditions.

With the basic steps having been defined in Sect. 1.3 and a formally sound mathematical basis of our approach, we can now go into more details of further prerequisites and the individual steps.

## 2.1 Long-term fatigue damage progression and energy production depending on external conditions and operational
planning

The standard approach in wind turbine design is the extrapolation of wind turbine loads from simulations to the design lifetime of e.g., 20 or 25 years. It is also a requirement for the certification of a turbine, defined in standards like (IEC, 2019; DNV GL, 2016). In the standards, design load cases (DLCs) determine the external conditions. To cover a wide range of sites, reference classes of wind conditions are defined, and conservative assumptions are often made. Currently, a fixed operational strategy 250 is assumed for each turbine. Therefore, there is currently no standardized process for the certification of wind farm control, e.g., by wake steering, since a subsequent adaptive adjustment of the turbine control is not covered by turbine certification standards. Integrating wind farm flow control into current and future certification procedures is part of ongoing work and discussions (Kölle et al., 2022). In the case of general adjustments to the operational strategy, such as through adjustments to the power curve or a power boost, evidence must be provided that the design damage budget is not exceeded. The major 255 difference between standard design calculations for fatigue damage and the presented approach for optimal planning is the





explicit integration of the control setpoints as a dependent variable on the external conditions, which can adaptively be selected and thus used as an optimization variable.

To cover the dependency of control on the external conditions, we assume that for each external input condition $x \in X$, there are one or multiple setpoints for the real-time controller $u(x) \in U$ that can be selected. The sets $X$ and $U$ will lateron
be clearly defined based on the system boundaries. The external input conditions are defined over a time increment $\Delta t$, and subsequently all of the dependent quantities are as well. It is at the time scale of the supervisory control loop, i.e., between minutes and hours. The idea now is to bin each operating condition separately, and then to determine the relative frequency for each combination of input conditions. The vector of input conditions is denoted as $x_j$ for a corresponding bin $j = 1, \ldots, B^x$, where $B^x$ is the total number of all bins of all input conditions.

The dimension of $x_j$ is given by the number of input conditions $w = Dim(X)$. The entire set of input conditions is denoted as $\bar{x} := \{x_j\}_{j=1}^{B^x}$. For each combination of input conditions, a separate operational strategy, i.e., setpoints of the system within the specified system boundaries, $\bar{u} := \{u(x_j)\}_{j=1}^{B^x}$ is defined. The total number of bins $B^x$ is usually defined as a fullfactorial multiplication $B^x = B^{x^{(1)}} \cdot \ldots \cdot B^{x^{(w)}}$, where $B^{x^{(i)}}$ denotes the number of bins defined for each condition $x^{(i)}$.

### 2.1.1 Use of frequency distribution for long-term prediction

In order to extrapolate the effects of the input conditions over long periods of time, it is usually made use of a relative frequency distribution $p_{\Delta\tau}$, which is representative of the input conditions within a period $\Delta\tau$. Hence

$$\sum_{j=1}^{B^x} p_{\Delta\tau}(x_j) = 1, \tag{4}$$

which can be scaled to an (absolute) frequency distribution

$$h_\tau(x) = p_{\Delta\tau} \cdot \tau \tag{5}$$

for a time period

$$\tau = Y\Delta\tau, Y \in \mathbb{N}. \tag{6}$$

For wind turbines, this usually assumes an annual distribution for the wind conditions, i.e., $\Delta\tau = 1\,\text{year}$, to be able to represent the variations through the different seasons. With the frequency distribution and the planned operational strategy, a mean damage increment $\Delta D_{fm}(\bar{u}, h_{\Delta\tau})$ can then be determined over the period $\Delta\tau$, i.e., an annual damage progress assuming
an annual wind distribution. The IEC-standard approach uses wind and turbulence classes for a type-certification of wind turbines. The wind speeds are modelled through a Rayleigh-distribution for the annual frequency, with a recommended binning resolution of 1 or 2 m/s for the wind speeds. The turbulence intensity is a fixed, but wind-speed dependent value using a 90% quantile (IEC, 2019). Principally, the choice of the frequency distribution is also part of the system boundaries to be selected and includes a large part of the uncertainties in the long-term planning together with the choice of the relevant external input
conditions. An alternative would be to use representative time series data combined with probabilistic approaches for the





fatigue damage prediction (Hübler, 2019). Such an approach requires more computational effort and is thus not yet suitable for a condition-based planning approach.

Using this and the assumption of a linear damage accumulation, damage can also be defined as a function of $\tau$ depending on the defined frequency distribution over that period and the operational strategy

$$D_{fm}(\tau;\bar{u},h_\tau) := \sum_{j=1}^{B^x} d_{fm}(x_j,u_j)h_\tau(x_j) = \underbrace{\sum_{j=1}^{B^x} d_{fm}(x_j,u_j)h_{\Delta\tau}(x_j)}_{\Delta D(\bar{u}),h_{\Delta\tau}} Y. \tag{7}$$

where $d_{fm}(x,u)$ is the damage rate under the external input conditions $x$ and the control setpoints $u$. It is also possible to compute the energy production accordingly by

$$E(\tau;\bar{u},h_\tau) := \sum_{j=1}^{B^x} P(x_j,u_j)h_\tau(x_j) = \underbrace{\sum_{j=1}^{B^x} P(x_j,u_j)h_{\Delta\tau}(x_j)}_{\Delta E(\bar{u}),h_{\Delta\tau}} Y. \tag{8}$$

where $P(x,u)$ is the power production under the input conditions and $\Delta E(\bar{u})$ the energy increment within $\Delta\tau$.

With adapted operational control for modified lifetime, the time period over which energy is produced is changed as well. The total lifetime energy yield can be computed by introducing a lifetime extension factor. It relates the lifetime with the reference operational strategy to the modified lifetime:

$$c^{ext} := \frac{\tau^{life}}{\tau^{ref}} = \frac{D_{fm}(\tau^{ref};\bar{u}^{ref},h_\tau)}{D_{fm}(\tau^{ref};\bar{u},h_\tau)} = \frac{1}{D_{fm}(\tau^{ref};\bar{u},h_\tau)} = \frac{\Delta D_{fm}(\bar{u}^{ref},h_{\Delta\tau})}{\Delta D_{fm}(\bar{u},h_{\Delta\tau})}. \tag{9}$$

Until now, the resulting lifetime was denoted as $\tau^{life}$, but in fact, this value is computed from damage $D_{fm}(\cdot)$ relevant for

a certain failure mode $fm$ and thus also only valid for this specific failure mode. For this reason, it is from now on denoted as $\tau_{fm}^{life}(\bar{u})$ and the extension factor as $c_{fm}^{ext}(\bar{u})$. With this, Eq. (9) can be expressed as

$$\tau_{fm}^{life}(\bar{u}) = \frac{1}{\Delta D_{fm}(\bar{u},h_{\Delta\tau})} = \frac{\Delta D_{fm}(\bar{u}^{ref},h_{\Delta\tau}) \cdot \tau^{ref}}{\Delta D_{fm}(\bar{u},h_{\Delta\tau})} = c_{fm}^{ext}(\bar{u}) \cdot \tau^{ref}. \tag{10}$$

The lifetime extension factor $c_{fm}^{ext}(\bar{u})$ can thus be used to compute the potential for lifetime extension on any time period where the damage increment is compared for two different strategies. Within the course of this work, $D_{fm}(\tau^{ref};\bar{u},h_\tau)$ is

computed in most cases, but it is important to realize that this value is actually closely related to the reference strategy. This becomes more clear when the damage rate is connected to the fatigue damage budget in the following Sect. 2.1.2.

Then, the energy production from the optimized operational strategy $\bar{u}^{opt}$ is given by

$$E\left(\tau_{fm}^{life}(\bar{u}^{opt});\bar{u}^{opt},h_\tau\right) = c_{fm}^{ext}(\bar{u}^{opt}) \cdot E\left(\tau^{ref};\bar{u}^{opt},h_\tau\right). \tag{11}$$

Up to now, the assumed damage progression is applicable to any failure mode where damage accumulates over time. With

this, we implicitly also assume that the details about material properties of the specific failure mode are included in $d_{fm}(x_j,u_j)$. Since fatigue is a design-driving failure mode and also one for which our proposed method is suited quite well, details for computing the damage rate for fatigue are explained in the following.





### 2.1.2 Damage and DEL calculation

Fatigue damage is induced by dynamic load cycles, which act due to forces and bending moments at various places of a
turbine. In simplified words, each load cycle leads to a small growth of tiny cracks inside of a material, which ultimately
causes the material to break or a component to fail. The crack appears at a very specific location, which is caused by the
dynamic loads at that point. Nevertheless, it is not possible to obtain the loads at every single location. Since the weakest
failure mode ultimately defines the failure of the system, it is often sufficient to consider only the most important loads at
specific locations for simplification. These loads are representatives for the fatigue damage of a material at a specific point like
e.g. at the blade root perpendicular to the rotor plane (Manwell et al., 2011). Within this section, the accumulation of damage
for arbitrary failure modes under various deterministic external conditions is explained and related to the standard approach
for counting load cycles and fatigue damage calculation. This is based on the assumption of a linear damage accumulation
by Palmgren and Miner (Miner, 1945). The high uncertainties resulting from this approach, especially for wind turbine rotor
blades but also for other components, is well known for a long time (Sutherland, 1999). In particular, it is not possible to
consider sequence effects of loads on the damage. Also, the material properties of composite materials can only be covered
to a small extent by this approach. As it was also mentioned in the literature study (Sect.1.1), estimating the state-of-health
of different wind turbine components is still a challenging task for feature research. Despite that, it remains common practice
to use linear damage accumulation for structural components of wind turbines. Especially for the comparison of loads under
different environmental conditions or control approaches, it remains a useful approach as a first step, before more advanced
evaluations can be examined with further development.

For the explanation of the general process, the failure mode index $fm$ is dropped. The fatigue damage rate of a time series
with input conditions $x_j$ is given by

$$d(x_j, u_j) = \sum_{i=1}^{n_{cyc,j}} \frac{n_{ij}}{N_{ij}} \tag{12}$$

for $i$ effective load collectives with a number of load cycles $n_i$. $N_i$ denotes the maximum bearable number of load cycles until
failure for the corresponding specific oscillation amplitude. The number of load cycles counted in the load time series of length
$\Delta t$ is denoted with $n_{cyc,j}$. The tolerable number of load cycles $N_{ij}$ depend on $D^{ult}$ and can be determined with

$$N_{ij} = \left( \frac{D^{ult}}{L_{ij}} \right)^m. \tag{13}$$

$L_{ij}$ represents the oscillation amplitude of a load cycle and are usually obtained from a rainflow counting algorithm. The
parameter $m$ is the component specific Wöhler exponent describing the slope of the S-N curve as negative inverse on a double
logarithmic axis. In the formulation of Eq. (13), the mean load is neglected and no Goodman correction is performed. The
value $D^{ult}$ denotes the ultimate design load which would lead to a damage of $D = 1$ if it occurred once. Therefore, $D^{ult}$
is a design parameter which needs to be determined from the design process under consideration of all conditions and their
frequency for the desired reference design period $\tau^{ref}$. In addition, it normally includes safety margins and design reserves.



For simplification $D^{ult}$ can be scaled in such way, that

345 $$D(\tau^{ref}; \bar{u}^{ref}, h^{ref}) = 1 \tag{14}$$

is valid, i.e. that fatigue damage is fully utilized with the reference operational strategy and under some site specific reference frequency distribution

$$h^{ref}(x) := h_{\tau^{ref}}(x) = p_{\Delta\tau}\tau^{ref}. \tag{15}$$

In this case, $D^{ult}$ can be expressed by making use of the damage equivalent fatigue load (DEL). It is a representative value 350 which would yield the same damage as the considered time varying signal with a constant amplitude and frequency. This value is referred to an equivalent number of load cycles $N^{eq}$. Then, the short-term DEL is computed by

$$DEL^{st}(x_j, u_j) = \left(\frac{\sum_i n_{ij}(L_{ij})^m}{N_{eq}}\right)^{\frac{1}{m}} \tag{16}$$

and the total DEL over the time span $\tau$ is given by

$$DEL(\tau; \bar{u}) = \left(\sum_{j=1}^{B^x} \left(DEL^{st}(x_j, u_j)\right)^m (h_\tau(x_j))\right)^{\frac{1}{m}}. \tag{17}$$

355 This can be used to solve Eq. (14) for $D^{ult}$:

$$
\begin{aligned}
1 &= \sum_{j=1}^{B^x} d(x_j, u_j^{ref}) h^{ref}(x_j) \\
&= \sum_{j=1}^{B^x} \sum_i^{n_{cyc,j}} \frac{n_{ij}}{N_{ij}} h^{ref}(x_j) \\
&= \sum_{j=1}^{B^x} \sum_i^{n_{cyc,j}} \frac{n_{ij}(L_{ij})^m}{(D^{ult})^m} h^{ref}(x_j) \\
&= \sum_{j=1}^{B^x} \underbrace{\left(DEL^{st}(x_j, u_j^{ref})\right)^m h^{ref}(x_j)}_{DEL(h^{ref}; \bar{u})^m} \frac{N_{eq}}{(D^{ult})^m}
\end{aligned}
$$

$$\Rightarrow D^{ult} = DEL(h^{ref}; \bar{u}^{ref})(N_{eq})^{\frac{1}{m}} = DEL^{ref}(N_{eq})^{\frac{1}{m}} \tag{18}$$

This can subsequently be inserted to Eq. (13) so that the damage can be expressed using the DELs as a relative value

$$d(x_j, u_j) = \sum_i^{n_{cyc,j}} \frac{n_{ij}}{N_{ij}} = \frac{n_{ij}(L_{ij})^m}{(DEL^{ref})^m(N_{eq})} = \frac{DEL^{st}(x_j, u_j)^m N_{eq}}{(DEL^{ref})^m N_{eq}} = \left(\frac{DEL^{st}(x_j, u_j)}{DEL^{ref}}\right)^m \tag{19}$$

In order to model the non-linear damage rate for the external conditions, surrogate models can be created by using the relationship to the short-term DELs which is given by equation (19). In principle, surrogate models for the damage rates could



directly be computed, but building up the models for the DEL is more common and easier to interpret because the Wöhler-exponent $m$ adds additional non-linearity to the damage value.

Having determined the basic relationships between all of the required quantities for the fatigue damage progression, the theoretical background for all of the four identified steps can be explained, starting with the setpoints for the real-time controller.

## 2.2    Step 1: Provide adaptable real-time controller of the wind turbine

In Sect. 2.1, the control setpoint $u(x)$ is introduced as an abstract value which can be selected based on the external input conditions $x$. This means, $u$ is basically a vector of different setpoints sent to the real-time controller, which in turn reacts by adjusting its own internal parameters. The exact effect that the setpoints have on system behavior strongly depends on the system at hand. Taking wake steering or reduction as an example, $u(x)$ would be the open-loop setpoint for the yaw angle or the amount of derating of the turbine depending on wind speed and wind direction (Nash et al., 2021). To discuss the way in

which $u$ can be integrated into an optimal planning approach, a little more background information on the control design of wind turbines is given, before methods for derating are presented controller setpoints of choice within this work.

For wind turbine controller design, the trade-off between various objectives needs to be found. Load reduction or mitigation of various components or specific failure modes competes with the influence of high control activity on actuators or other failure modes. The main objectives remain reliably producing energy while at the same time meeting the requirements of a grid

operator (Burton et al., 2011; Njiri and Söffker, 2016; Requate et al., 2020). Due to the varying influence of external conditions and the aero-servo-hydro-dynamic coupling of all the failure modes, the selection of a trade-off depends on multiple individual factors for each turbine. When aiming at adaptive operation, multiple control setpoints need to be selected so that each of them prioritizes a different objective or a combination of such. Due to the strong influence of the external conditions (mainly wind conditions) on the wind turbine performance, an individual trade-off for each of the various conditions might actually be

required.

Therefore, providing an adaptable real-time controller definitely possible in many ways for wind turbines. Through a preselection of control setpoints, one can reduce the computational effort and use them for simulations to built up surrogate models. Since derating can also be applied to any existing turbine and has the ability to reduce fatigue damage, it is selected as an instrument for the planning of damage progression in this work. Also, the conflict to the energy production of the individual

turbine is directly clear. The derating of a turbine can be performed in several ways, and the specific method will have a large influence on the load reduction. There are several studies which investigate derating methods with respect to their influence on various objectives like power regulation for the grid (ancillary services), wake reduction (power maximization) or loads. In Houck (2022), several studies on derating (or axial-induction control) are summarized and sorted into the mentioned categories. Many studies investigate load reduction as a side effect, while the main objective is either the power regulation or reducing

the wake on the downstream turbine. For the discussion of derating methods, one has to distinguish between the partial and the full load region of a wind turbine controller. Below the rated wind speed, the dominant goal in normal operation is to obtain the maximum available power from the wind. In the full load region, the generator speed is held constant by pitching the blades and the turbine operates at its rated generator torque and power. The partial load region, where maximum power



point tracking (MPPT) is applied, is also referred to as region 2 and the full-load region above rated wind speed as region 3.
The MPPT in partial load is obtained by operating the turbine at the maximum value of the power coefficient $c_p$, which is a function of the tip-speed ratio $\lambda$ and the pitch angle $\beta$. Therefore, the power can be reduced by adapting either of them or both. Benefits and shortcomings of different approaches have been applied and investigated in various studies (Zhu et al., 2017; Astrain Juangarcia et al., 2018; Lio et al., 2018; Meng et al., 2020). For the reduction of wake effects, minimizing the thrust is beneficial, but it can have negative effects on tower loads when the rotor speed reduction leads to resonance effects (Meng et al., 2020). In Astrain Juangarcia et al. (2018) similar load reducing effects were observed when the tip speed ratio was held constant (so called constant-$\lambda$-method) and when thrust was minimized. Overall, the studies show that the setpoints need to be selected specifically for each turbine and objective.

In the full load region, the torque set point is normally reduced for derating. It allows for a fast recovery of power when derating is no longer required, and is thus beneficial for ancillary services (Fleming et al., 2016; van der Hoek et al., 2018). However, it only has a minor effect on the fatigue loads of the blade and the tower. Reducing the generator speed mainly has a strong positive effect on the blade loads in flapwise direction (Requate and Meyer, 2020), while reducing the torque has a positive influence on the driving torque loads (Pettas and Cheng, 2018). The effect on the tower loads are quite turbine dependent because a reduction in generator speed can reduce oscillations to some extent but often also increases them due to the lowered aerodynamic damping (van der Hoek et al., 2018). Therefore, a mixed method between reducing torque and speed might be advantageous, again depending on individual objectives and turbine characteristics.

Concluding this step, it is evident that adaptable real-time controllers already exist for wind turbines, but choice of a specific method is crucial for the use under system boundaries of the planning. In Sect. 4.1, a specific method is implemented for the application example. The setpoint is then used as one of the inputs for a surrogate model.

## 2.3 Step 2: Build surrogate models for damage progression and energy production

In order to cover the influence of the various influences of external conditons and control on the fatigue damage, surrogate models have gained growing research interest. The approaches for surrogate modelling, sometimes also called meta-modelling, have in common that aero-elastic simulations are used to create a database of fatigue loads for various input conditions. Subsequently, a mathematical model is trained to represent the complex relationship of inputs to the fatigue of different load representatives with a simplified mathematical model. Usually, the short-term damage equivalent loads of different load signals and the electrical power are considered as output quantities as these are most commonly used for fatigue design calculations because it allows to integrate the influence of various external conditions in the design process. A good overview of different surrogate methods and a comparison of their performance is given in Dimitrov et al. (2018). Apart from linear interpolation or polynomial regression, the most popular methods, which are being used for the investigation of surrogates, are gaussian regression (often referred to as Kriging), polynomial chaos expansion and artificial neural networks (Dimitrov, 2019; Hübler, 2019; Slot et al., 2020; Gasparis et al., 2020; Debusscher et al., 2022; Singh et al., 2022). Depending on the purpose, a trade-off between accuracy, interpretability, training time, number of required samples and also evaluation time needs to be found, and a method is selected accordingly. All of them need to deal with the uncertainties due to the stochastic nature of the wind, which





is also reflected in the fatigue damage and the DELs. Bossanyi (2022) uses a different approach for creating the surrogates by splitting up the inputs into deterministic and stochastic influences.

To be able to use the short-term damage for optimization, it is required to apply a surrogate model because the damage rate needs to be evaluated with a low computational time for an arbitrary combination of inputs. In principle, any kind of surrogate function $f_{DEL_{fm}}(z)$ for each considered failure mode $fm$ and for the power production $f_P(z)$ needs to be found to determine the relationship for an arbitrary input $z = (x, u(x))$. The fit is based on short-term DELs computed for a set of input samples of the external conditions and the control setpoints which are denoted as $\hat{z} = (\hat{x}, \hat{u}(x))$. The value $DEL_{fm}^{st}(\hat{z})$ is obtained

through aero-elastic simulations of the wind turbine model and a subsequent evaluation using the rainflow counting algorithm and Eq. (16). Also, the mean power is computed $P(\hat{z})$.

    To account for the randomness in the incoming wind and offshore also wave conditions, various realizations of the same mean input characteristics are usually simulated. Those are determined through pseudo-random seeds. Therefore, the total simulation time is the time of a single aero-elastic simulation times the number of seeds. The outputs of the simulation are always scaled

so that they match the corresponding time span $\Delta t$ for the damage rate which will be used to compute the damage progression. The sampling of $\hat{z}$ needs to be chosen so that it matches with the surrogate modelling approach. Classical interpolation usually requires fullfactorial sampling of $\hat{z}$ so that the number of samples increases exponentially with the dimension of $z$, i.e. the number of input variables, and thus leads to a high number of computationally extensive load simulations. For a low number of inputs, however, a fullfactorial sampling allows for a deeper understanding of the results in every combination. It also allows

for interpolation, which has a major advantage of a straight-forward implementation and no need for tuning of hyperparameters of the surrogate model. Due to the randomness in the input conditions, which is reflected to varying degrees in the behavior of the DELs, interpolation is often not a suitable approach. Additional requirements result from the necessary suitability of the surrogate model for the application in optimisation. For gradient-based optimization, differentiability of the surrogate model is required or at least a sufficient smoothness for computing the difference quotient. One major input which also influences the

choice and applicability of the surrogate model is the control setpoints. The surrogate needs to be able to cover the influence of the control setpoints on the loads in a deterministic way, despite the stochastic influence of the wind. A specific surrogate model for the application examples is selected in Sect. 4.2. With the surrogate model computing the damage rate, the optimization problem can be built for the operation planning.

## 2.4   Step 3: Determine optimal condition-based operational strategies for lifetime planning

The central part of this work is to build up the optimization problem which allows creating an optimal planning of the damage progression over a long period of time or the entire lifetime respectively. The key is to formulate the problem in such way, that a control setpoint is found for each external condition while long-term objectives are fulfilled. Neglecting economic factors and other restrictions at first, it is ecologically most beneficial to get the maximum amount of energy over the lifetime $\tau^{life}$ of the turbine while the fatigue budget of each component is fully used up. Within this work, we set a single turbine and

its surrounding external influencing conditions as the system boundary and the mathematical formulation of this problem is presented in a general form. More details on the system boundaries of the application example are provided in Sect. 3.





The operational strategy $\bar{u} = \{u(x_j)\}_{j=1}^{B^x}$ is optimized for each of the external conditions which were previously selected by the definition of the system boundaries. It follows, that the number of selected independent control setpoints, defined by $Dim(U)$ and the number of bins which are used for the external conditions $B^x$, determine the number of optimization variabales, which is equal to $B^x \cdot Dim(U)$. Within the scope of this work $Dim(U)$ is equal to 1 because a single derating strategy will be applied. With a fixed known target fatigue budget $D_{fm}^{target}$ for failure mode $fm \in \mathcal{F}$, the optimization problem is formulated as

$$\max_{\bar{u}} \sum_{j=1}^{B^x} P(x_j, u(x_j)) h^{ref}(x_j)$$

$$\text{subset to } \sum_{j=1}^{B^x} d_{fm}(x_j, u(x_j)) h^{ref}(x_j) \leq D_{fm}^{target}, \; \forall fm \in \mathcal{F}. \tag{20}$$

Using this simple and compact formulation, it is possible to spare the fatigue budget when the damage rate is high compared to power production. When the damage of all failure is reduced compared to a baseline operation $\bar{u}^{ref}$, i.e. $D_{fm}^{target} \leq D_{fm}^{ref}$, the turbine can be operated for a longer time and ultimately more energy can be produced.

Clearly, the solution strongly depends on the selected failure modes, their behaviour of the damage rate determined from the surrogate models and on the target fatigue budget. When several failure modes should be optimized simultaneously, it might be impossible to fulfill the constraints and no solution can be found. Therefore, the selection of the target budget strongly depends on the specific problem which is individual for a specific wind farm or wind turbine respectively. On the one hand, many other factors need to be considered in order to decide when and how a component should be allowed to fail, depending on the relevant failure modes. A major factor are component costs, which will determine if a failure leads to a decommissioning of the turbine or a simple exchange, which needs to be aligned with maintenance planning. On the other hand, the formulation in (20) provides a clear separation of the technical aspect from the economic aspect and therefore allows investigating the relationship between damage progression and energy production for different components under consideration of the operational strategies over long periods of time. The formulation of (20) can also directly be used to create the Pareto-front between damage and energy production by principally applying the Epsilon-constraint method for multi-objective optimization (Chiandussi et al., 2012), i.e. by fixing various combinations of the target values $D_{fm}^{target}$. This approach is applied in Sect. 4.3.2. Some economic aspects which arise from a financing model for debt repayment are covered in the fourth step, which is applied in Sect. 4.4. The underlying financing model is explained in the next section.

## 2.5 Step 4: Select economically best operational lifetime-planning strategy

With the presented optimal planning approach, higher total energy yield can only be achieved with lifetime extension, which in turn is only made possible by accepting lower yearly energy production throughout the lifetime. With the approach presented in Sect. 2.4, more energy can be obtained from the utilized materials, but the economic impact is neglected.

A first evaluation of the effect of optimized operational planning on total revenue is possible with a simplified approach that includes a basic financing model. This is based on the net present value $NPV$, which maps a future payment to its current





value. We assume a constant interest rate $C_{WACC}$ covered by the weighted average costs of capital (WACC), constant annual maintenance costs $C_{OPEX}$ and a constant average price of electricity $C_{elPrice}$. The repayment can be variable over the entire operating period and is made depending on the annual energy yield $E(\Delta\tau; \bar{u}, h_{\Delta\tau})$. Currently, it is assumed to be constant in each year, because we use the same operational strategy and frequency distribution. With these parameters, the NPV can be computed for payments until a given year $Y$:

$$NPV(Y) = \sum_{t=0}^{Y} \frac{C_{elPrice} \cdot E(\Delta\tau; \bar{u}, h_{\Delta\tau}) - C_{Opex}}{(1 + C_{WACC})^t}, \tag{21}$$

The future value at the end of the lifetime of an adapted operating strategy is given by $NPV(Y^{life})$ where the number of full operating years with the strategy is defined as

$$Y^{life} := \left\lfloor \frac{\tau_{fm}^{life}(\bar{u})}{\Delta\tau} \right\rfloor. \tag{22}$$

This model maps all future payments to their current value, and thus also gives an upper bound to the initial investment that is permissible. Any revenue above the initial investment leads to additional profit.

## 3   System definition for application example: One turbine in a small regular-grid wind farm

The presented method for optimal planning is applied to the demonstration example. Before the four steps can be applied, the system boundaries need to be determined.

In our application example, we want to focus on optimal operation of a single turbine within a wind farm. This means, that although our system boundary is well-defined around the single turbine, effects from the surrounding wind farm have to be taken into account as well. These include mainly the wake effects from other turbines, which act on the considered turbine and are, under normal operation, a main driver of its loads. Each single considered turbine will thus be able to react to the wake effects from the surrounding turbines, but the effect from changes in control on the wake cannot be considered yet. How to address this issue and other shortcomings and potential improvements of the approach will be discussed in Sect. 5.

The example wind farm consists of 9 turbines with a regular $3 \times 3$ layout, shown in figure 3. The turbine spacing is 8 rotor diameters in x direction and 4 rotor diameters in y direction. This example farm was already used in Schmidt et al. (2021). The generic direct-drive wind turbine IWT7.5 with a nominal power of 7.5 MW, rotor diameter of 164 m and a hub height of 100 m is used (Popko et al., 2018).

### 3.1   System boundaries for considered system: Modelling of single turbine

The system boundaries for the single turbine are defined by the selected external input conditions, i.e. the definition of $X$, the utilized aero-elastic model for the simulation of loads. In addition, the failure modes need to be selected and the damage rates with their specific parameters need to be computed from the loads.





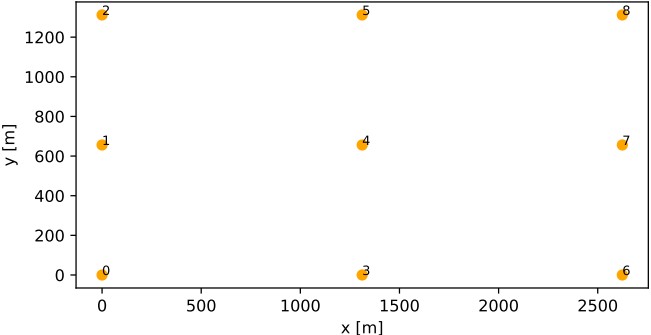

**Figure 3.** Layout of example wind farm

### 3.1.1 Computation of turbine loads

To compute the loads of the turbine, the aero-elastic load simulation tool "The Modelica library for Wind Turbines" (MoWiT) (Thomas, 2022) is employed. Three-dimensional wind fields covering the properties of the external conditions within for simulation is used as input. They are created with the software Turbsim (Jonkman, 2009).

### 3.1.2 Selected external environmental input conditions

For the application of optimal planning in this paper, two major environmental inputs influencing the wind turbine loads in power production mode are considered as local input conditions: mean wind speed $v$ and turbulence intensity at hub height $TI$. Those input conditions are defined locally as the inflow to a single turbine which positioned its rotor perpendicular to the main inflow wind direction. The local inflow on a turbine from wake effects is covered through an increase in turbulence

intensity only. This simplification allows splitting the aero-elastic turbine simulations from the wake modelling, and thus reduces simulation effort. The range of simulated data will be defined in Sect. 2.3 where the surrogate models for the example is built up.

### 3.1.3 Selected failure modes

For the demonstration of the approach, the structural loads of the blades and the tower are considered. Both are supposed to

last for the complete design lifetime of 20 or 25 years. Both are also influenced by the turbine controller and the wake induced turbulence. For the blades, the flapwise and edgewise bending moments are considered as separate failure modes, because they represent the two major load driving moments on the rotor blades. The distribution of moments along the blade root is not fully reflected by this approach, but it is possible to cover the differing effects which influence both moments. While the edgewise bm is mainly driven by the gravitational loads of the rotating system, the flapwise bm is strongly influenced by the

turbulence and thus from the wake effects. For the tower, the combined bending moment at the bottom is utilized as failure mode. This is dominated by the fore-aft movement of the tower. The direction of the fore-aft moment actually depends on the





wind direction, since the coordinate system is always defined with respect to the rotor plane perpendicular to the main wind direction. Therefore, the use of the combined moment also neglects some influencing factors, but still represents the influence on the tower fatigue loads sufficiently well for demonstration.

The considered loads, their corresponding abbreviations and the utilized Wöhler-exponent $m$ are summarized in Table 1. For the blades, an exponent of $m = 10$ is selected, which is a common choice for representing the blade root laminate. Using linear fatigue accumulation by using DEL is a very strong simplification for the fatigue degradation of laminate, which is a composite material containing fibre glass. Using this approach is still standard for design calculation and allows for a straight forward use without detailed knowledge about the material properties. For the tower, an exponent of $m = 3$ is used, which is

representative for steel components and $m = 10$ for the blade loads as an approximate for fibre (Sutherland, 1999). In total, the three selected failure modes are representatives for the fatigue accumulation for different components in the wind turbine and therefore applicable for the demonstration of the optimal planning approach.

**Table 1.** Summary of terms for the selected failure modes

| Load | Flapwise Bending Moment | Edgewise Bending Moment | Tower Bottom Bending Moment |
|---|---|---|---|
| Abbreviation | Flapwise bm | Edgewise bm | Tower (bottom) bm |
| Wöhler Exponent | 10 | 10 | 3 |
| Short-Term DEL | $DEL^{st}_{flap}(x,u(x))$ [Nm] | $DEL^{st}_{edge}(x,u(x))$ [Nm] | $DEL^{st}_{tower}(x,u(x))$ [Nm] |
| Damage Rate | $d_{flap}(x,u(x))$ [1/h] | $d_{edge}(x,u(x))$ [1/h] | $d_{tower}(x,u(x))$ [1/h] |

### 3.2 From surrounding system to considered system: From wind farm to turbine

Since the frequency distribution of the wind direction has a major influence on how often a turbine is affected by the wake

of other turbine, it needs to be considered for an optimal planning of damage. Therefore, a site specific wind distribution is required to cover the influences of the surrounding system. The global wind conditions need to be transferred to local input conditions for the considered system, i.e. the single wind turbine. This is done by using a basic steady-state wake model, which transfers from ambient wind speed and wind direction to local wind speed and TI. At first, the ambient wind data is defined before the wake model is briefly explained and a local wind distribution is derived from both.

#### 3.2.1 Global wind conditions (Ambient Wind Data)

ERA5-data representative of a wind farm in the North Sea are used (Hersbach et al., 2018). From the time series, a 30-year period from 1990 to 2019 with a resolution of 1h, the mean wind speed and wind direction at 100 m height is extracted to create a relative frequency distribution of ambient wind speed $\bar{v}^{amb}$ and wind direction $\bar{\theta}^{amb}$ which are both subdivided into bins. This relative distribution is representative for a one-year period. Therefore, the reference relative wind distribution for the

ambient wind conditions is $\mathbf{p}^{ref}_{\Delta\tau}(\bar{v}^{amb}, \bar{\theta}^{amb})$ with $\Delta\tau = 1year$. Because wind speed and direction are covered separately, the total number of bins $B^x$ is subdivided into bins for each direction. The wind speeds $\bar{v}^{amb}$ are first binned with a resolution of 1



m/s from 1.5 to 49.5 m/s. Afterwards, the values below 4.5 m/s and above 23.5 m/s are removed (Number of wind speed bins $B^{v^{amb}} = 20$). This implies that the turbine is assumed to be in idling mode below and above these values. Therefore, values between these two wind speeds, where derating actually does have an influence on the turbine, are considered for optimization.

It also means that $\mathbf{p}_{\Delta\tau}^{ref}(\bar{v}^{amb}, \bar{\theta}^{amb})$ does not sum up to 1 anymore, but to 0.9. The wind direction is binned with a resolution of $2°$ from $0°$ to $358°$ (Number of wind speed bins $B^{\theta^{amb}} = 180$). This results in a total number of $B^{x^{amb}} = 180 \cdot 20 = 3600$ bins. The ambient turbulence intensity is not available from the ERA5-data. It is modelled as a fixed value, and is set to 5% for all wind speeds and wind directions. Within the system boundaries of the application example, we concentrate on the additional TI from wake. Other external inputs like e.g. atmospheric turbulence or temperature are outside the system boundaries and

implicitly considered as fixed values.

### 3.2.2 Wake modelling

To compute the wake effects, the IWES software FOXES[3] is used. More details on the software can be found in (Schmidt et al., 2021). In this case, only a small part of the software is used to compute the steady-state wake effects for each of the turbines under the reference conditions. From the view of a single turbine, a function which maps the ambient mean

wind speed $v^{amb}$ and wind direction $\theta^{amb}$ to the local mean wind speed $v$ and turbulence intensity $TI$ is required. From the view of a specific turbine, the wake modelling function depends on number, distance and also operation of the surrounding turbines. Thus, the wake function is valid separately for each turbine $s = \{1, \dots, S\}$ in a wind farm with $S$ turbines and denoted as $f_s^{wake}(v^{amb}, \theta^{amb})$. The local wind speed is computed using the Gauss-type wake model by Bastankhah and Porté-Agel (2016). The wake induced turbulence intensity is calculated using the top-hat wake model as described in IEC (2019).

### 3.2.3 Local frequency distribution

For computing the damage and energy production with the frequency distribution of the ambient wind conditions defined above, the wake modelling function needs to be included in equations (7) and (8). For clarification, the sum over $B^x$ which was formerly used is now subdivided into two separate sums over the specifically defined input conditions. This implies for a turbine $s$:

$$D_{fm}(\tau; \bar{u}) = \sum_{j=1}^{B^{v^{amb}}} \sum_{i=1}^{B^{\theta}} d_{fm}\left(f_s^{wake}(v_j^{amb}, \theta_i^{amb}), u((v_j^{amb}, \theta_i^{amb}))\right) h^{ref}(v_j^{amb}, \theta_i^{amb}) \tag{23}$$

and

$$E(\tau; \bar{u}) = \sum_{j=1}^{B^{v^{amb}}} \sum_{i=1}^{B^{\theta}} P\left(f_s^{wake}(v_j^{amb}, \theta_i^{amb}), u((v_j^{amb}, \theta_i^{amb}))\right) h^{ref}(v_j^{amb}, \theta_i^{amb}). \tag{24}$$

As explained in Sect. 2.4, the number of external wind condition bins determine the number of optimization variables. Reducing those can thus reduce the effort for optimization. Since the interaction of the turbines is only modelled unidirectional,

---

[3]The software was formerly named flappy, and version v0.4.3.3 was applied.



without considering the influence of the changing control setpoint on the wake of other turbines, it is possible to create a local frequency distribution for each turbine, which only depends on the distribution of local wind speeds and turbulence. To do so, the frequencies of $\mathbf{p}_{\Delta\tau}^{ref}(v^{amb}, \theta^{amb})$ are binned again into $B^v = 20$ wind bins, as before, and $B^{TI} = 25$ TI bins with a width of 1% starting from 5%, resulting in 500 total bins.D The frequency distribution for the additional binning is denoted as $\tilde{h}_s^{ref}$, and is only valid separately for each turbine $s$. Then, the damage and energy calculation from equations (23) and (24) can be simplified to

$$\tilde{D}_{fm}(\tau; \bar{u}) = \sum_{j=1}^{B^v} \sum_{i=1}^{B^{TI}} d_{fm}(v_j, TI_i, u(v_j, TI_i)) \tilde{h}_s^{ref}(v_j, TI_i). \tag{25}$$

and

$$\tilde{E}(\tau; \bar{u}) = \sum_{j=1}^{B^v} \sum_{i=1}^{B^{TI}} P(v_j, TI_i, u(v_j, TI_i)) \tilde{h}_s^{ref}(v_j, TI_i). \tag{26}$$

This simplified form will thus be used in the results part where the approach is applied. From the additional binning, additional uncertainty is added to the results. This is however neglectably small. It is also possible to use equations (23) and (24) directly for the optimization, with a higher computational time.

## 4 Application of four-step method

With the specified system boundaries, the four-step procedure described in Sect. 1.3 can be applied to the demonstration example. At first, the specific choice for the derating method is explained. Afterwards, the results of the load surrogate models are presented, mainly to understand the influence of the control setpoints under the input conditions. Afterwards, the optimization problem is conducted for two different use cases pursuing different goals. At first, levelling the damage of all turbines of the artificial wind farm is intended. Secondly, the trade-off between annual damage and energy production is found separately for each of the selected failure modes in the form of Pareto-fronts.

### 4.1 Step 1: Provide adaptable real-time controller of the wind turbine

While derating was already defined as suitable choice for load reduction, the implementation of specific setpoints of the real-time controller still needs to be specified. The IWT7.5 is controlled with the IWES research controller developed internally in Matlab Simulink (Wiens, 2021). It implements a standard torque-pitch controller of a wind turbine (see e.g. Burton et al. (2011) or Njiri and Söffker (2016)). In region 2, the $k\omega^2$-law is applied to set the generator torque $M$. The coefficient $k$ is derived from the aerodynamic characteristics of the turbine, with the standard approach to achieve MPPT. In region 3 the turbine operates at its rated power $P_r$. Here, a PI-controller for keeping the rated generator speed $\omega_r$ constant by pitching the blades is implemented. The generator torque is fixed to its rated value $M_r$. The transition between region 2 and 3 is simply determined by a linear relationship between torque and speed, which starts from 95 % of $\omega_r$.





Within the sytem boundaries of this study, the main objective is not to determine the best fitting derating method for the generic wind turbine, but to show the benefits of using derating for an optimal planning. Therefore, the choice was conducted

based on the findings from the cited literature and from previous experience with the IWT7.5 and not through an extensive study and tuning of the controller under various conditions. Also, no additional features like individual pitch control or active dampers are activated. For a real-world application, fine-tuning the controller for every derating configuration would be beneficial and could lead to an improved performance with respect to loads and power. For showing the benefits of the optimal planning approach, such a fine-tuning is not yet required. The presented derating method represents one potential choice for

the derating of a turbine by a proportional factor in all regions, and thus a trade-off between the induced fatigue damage and power production.

The derating method is thus implemented such that it reduces power in partial and in full load by a percentage factor. Such a derating method is referred to as proportional delta control in Elorza et al. (2019) or percentage reserve in van der Hoek et al. (2018). By using a single derating factor $u(x) = \delta_P$ as a setpoint for the real-time controller, the combination of a derating

method with the creation of surrogate models and the use for the nonlinear optimization is facilitated. It is defined as percentage power factor $\delta_P \in [\delta_P^{min}, 100 \ \%]$. The structural fatigue loads of both, the blades and on the tower should be reduced. A derating down to 50 % is considered as sufficient for this purpose, i.e. $\delta_P^{min} = 50 \ \%$.

In full load operation, the main goal is to reduce the fatigue loads of the flapwise bm while not increasing or slightly decreasing the tower bm fatigue. In partial load, both tower and blade fatigue loads should be decreased. To do so, the so-called

constant-$\lambda$ method is implemented (Astrain Juangarcia et al., 2018). This is achieved by finding the steady-state pitch angle $\beta$ so that the reduced power coefficient $\delta_P c_p$ is found while $\lambda$ is kept constant. From these values, the coefficient $k^d$ for the derated operation can again be computed.

In region 3, a combined method for reducing the rated generator torque $M_r$ and $\omega_r$ is selected. At first, the percentage power reduction factor $\delta_P \in [\delta_P^{min}, 100 \ \%]$ is defined to compute the reduced rated power by

$$P_r^d = \delta_P P_r. \tag{27}$$

The reduction factor for the rated generator speed $\delta_\omega \in [\delta_\omega^{min}, 100 \ \%]$ is then defined with a quadratic relationship to the reduction in power by

$$\delta_\omega = \delta_\omega^{min} + (1 - \delta_\omega^{min}) \left( \frac{\delta_P - \delta_P^{min}}{1 - \delta_P^{min}} \right)^2 \tag{28}$$

so that the derated generator speed is given by

$$\omega_r^d = \delta_\omega \omega_r \tag{29}$$

and the generator torque by

$$M_r^d = \frac{\delta_P}{\delta_\omega} M_r. \tag{30}$$





In this case, we use $\delta_\omega^{min} = 80$ %. Using this approach, we aim at strongly reducing fatigue loads on the flapwise bm through reduction in generator speed while still providing sufficient aero-dynamic damping on the tower. For the transition

region between region 2 and region 3, the slope of the linear transition function was kept constant.

In Fig. 4, the operating points of the controller for the selected setpoints are presented. Figure 4a shows the speed-torque curve of the controller. The end point of the curves always determines the combination of $M_r$ and $\omega_r$ so that the process of these point reflects the relationship from Eq. (28) to Eq. (30) in region 3. In region 2, the constant-$\lambda$ strategy determines a specific combination of the factor $k_{ctrl}^d$ and the static pitch angle. The value for $k^d$ is reduced with decreasing $\delta_P$, as it can be seen

from the quadratic part of the speed-torque-curves in Fig. 4a. The steady-state operating points of the pitch-angle are plotted over the wind speed in Fig. 4b. The pitch angle in region 2 is kept constant below the rated wind speed for all of the setpoints for the percantage power. The constant value increases with decreasing $\delta_P$. Above rated wind peed, the change of the derived setpoints for the rated generator speed and torque also results in an increasing steady-state pitch angle of the PI-controller. In combination, this results in the steady-state power curves which are shown in Fig. 4c.

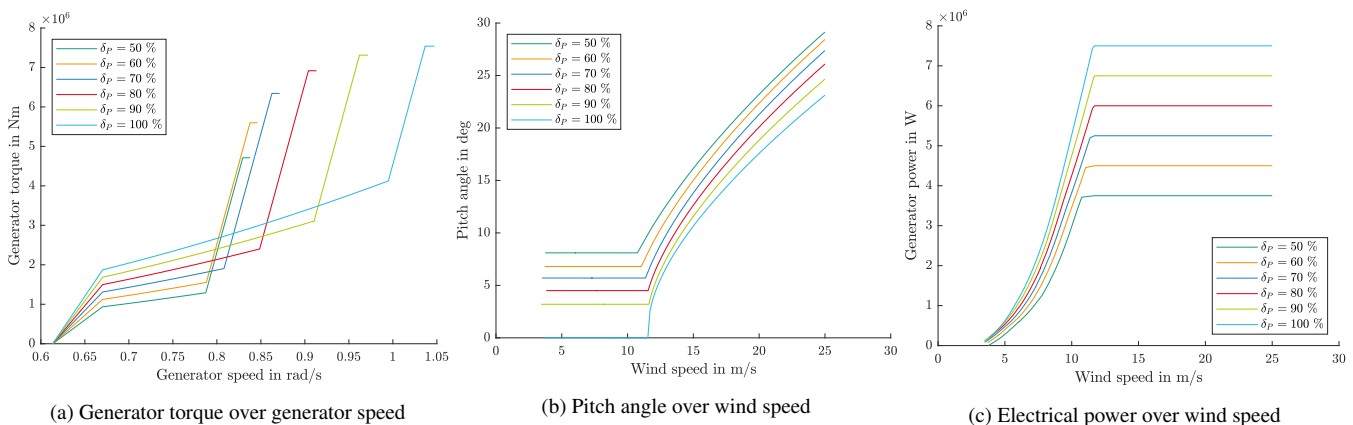

(a) Generator torque over generator speed     (b) Pitch angle over wind speed     (c) Electrical power over wind speed

**Figure 4.** Operating points of the real-time controller for selected derating methods

In order to study the controller under various conditions, the evaluation of aero-elastic load simulations is required. Those load simulations are also used as an input for the surrogate models. Therefore, the selected approach and the results of surrogate models are presented first before the influence of the control setpoints on the DELs of the selected failure modes are discussed.

**4.2   Step 2: Build surrogate models for damage progression and energy production**

The surrogate models for optimal planning need to be able to represent the relation of the derating setpoint to load reduction

for each external input condition adequately. In addition, a deterministic differentiable function is helpful, if not even required, to be used for optimization. Due to the high number of optimization variables, gradient-based optimization is favoured over heuristic optimization approaches, which makes differentiability a requirement for convergence criteria. Within this work, the main purpose of the surrogates model is to use it as an instrument for showing the long-term optimal planning approach and not finding a perfectly suiting surrogate for this purpose yet. Finding a surrogate for any intermediate data point also means,





that any intermediate value of percentage power between $\delta_P = [50\%, 100\%]$ can be selected. Using the approach, which is presented in 4.1, a continuous selection of derating would be theoretically possible by providing the according setpoints in partial and full-load. If a continuous derating or a discrete selection of control methods is advantageous or feasible for a real-world application will depend on the use specific wind turbine controller and use case. In order to model the relationship between percentage power reduction and induced damage for optimization, a continuous surrogate model is advantageous and

the discrete selection of a setpoint can be implemented on the operating stage.

      Multidimensional polynomial regression models are selected as surrogate models for the DELs due to their simple usage, their differentiability and their fast training and evaluation time. For the electrical power, a linear interpolation is used. Replacing the selected methods by other models which were described in Sect. 2.3 will not affect their application for this purpose if suitability for optimization and accuracy is sufficiently tested.

To obtain the parameters of the polynomial regression model for the DELs, a least squares approach

$$\min_q \| f_{DEL_k}(\hat{z}; q) - DEL^{st}(\hat{z}), \| \tag{31}$$

is used. The vector $q$ denotes the coefficients of the polynomial, where the maximum degree of 5 is found through the validation error from a cross-validation process. In this case, $DEL^{st}(\hat{z})$ are the results obtained from aero-elastic load simulations within the system boundaries defined in Sect. 3. 6 10-minute simulations with 6 different pseudo-random seeds of the wind fields are

performed, and the outputs are obtained at the time scale $\Delta t = 1\ h$. A fullfactorial sampling for the local input conditions $x$ together with the percentage power is defined, i.e. $u(x) = \delta_P(x)$. The sampling values are provided in Table 2.

**Table 2.** Input sampling for load simulations

| Wind Speed $\hat{v}$ | Turbulence intensity $\widehat{TI}$ | Percentage power $\hat{\delta}_P$ |
|---|---|---|
| 4,5,…,25 m/s | $(\sqrt{2})^i \% \forall i = 2, \ldots, 11$ | 50,60,…, 100 % |

      The short-term DELs are obtained using the rainflow counting algorithm as described in Sect. 2.1.2 for the selected failure modes (See Sect. 3.1.3). For the further usage, i.e. the computation of the damage rates being used for optimization, the surrogate models $DEL_k^{st}(x, u(x)) = f_{DEL_k}(x, u(x))$ are used and transferred to a damage rate by Eq. (19). The linear interpolation

of power $P(x, u(x)) = f_P(x, u(x))$ is directly used as mean value over the time span $\Delta t$.

      Within this section, two things are presented and discussed. On the one hand, the accuracy of the surrogate fit on the data needs to be evaluated. On the other hand, it is important to understand the relationship between external input condition, control setpoint and DELs, to be able to assess the selected control methods and also to interpret the long-term planning results. The first point can be examined by looking at the overall error between simulated data and the surrogate model first, but also by a

detailed assessment of the surrogate for certain input combinations. The latter can also be used, to address the second point, i.e. for an interpretation of the simulated behaviour.

      At first, the output of the surrogate model is plotted against the simulated data in Fig. 5 and the relative absolute error on the data (training error) is computed. Afterwards, one of three inputs is fixed and the evaluated surrogate model (solid lines) as





well as the simulated training data (dots in same colour as solid line) are shown exemplary for the other two inputs in Fig. 6 and
Fig. 7. For all three failure modes, a general agreement of the surrogate model to the data can be observed in all of the figures.
For the flapwise bm and the tower bm, the error is significantly higher than for the edgewise bm (see figure titles). This can also
be observed by the higher spread of the data points for these two failure modes in Fig. 5 and the higher deviation of datapoints
to the surrogate fit in the other plots. This is mainly caused from the higher dependency on the wind turbulence, which also
creates more variance in the DELs. Especially for the tower, the high variation in the simulation data makes it difficult to create
a surrogate model. One can see from Fig. 6 and Fig. 7 that the models performs well in some regions but worse in others. The
error could be reduced by creating a higher number of simulations for training, either with more samples or with more random
seeds. In addition, a different approach for the surrogate models like gaussian regression or neural networks as mentioned in
Sect. 2.3 could be applied. Overall, the models fit sufficiently well for demonstrating the optimal planning approach, while
further improvements are certainly possible and also required for a more accurate result.

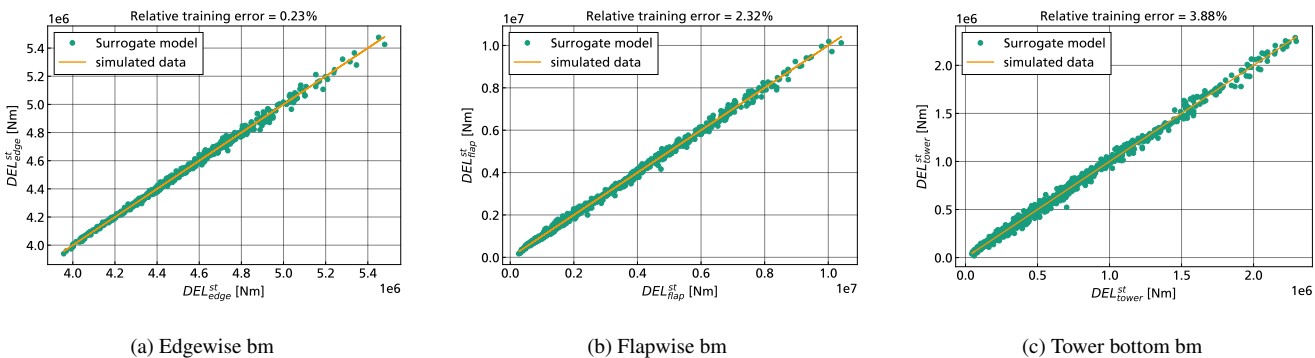

| (a) Edgewise bm | (b) Flapwise bm | (c) Tower bottom bm |

**Figure 5.** Error in training data for polynomial regression surrogate models

The reason for the low accuracy of the surrogate model on the DELs of the tower bm is also caused by the varying influence
of the control setpoints on the DELs. By taking a closer look on Fig. 6 and Fig. 7, such details about the behaviour of the
DELs can be discussed. Figure 6 shows the results with the wind speed on the x-axis for different values of percentage power
with a fix $TI = 11.3$ %. All values are scaled by the output $v = 11\ m/s$, $TI = 11.3$ % and $\delta_P$=100 % so that the relative
behaviour of the DEL and the power can be seen. The most important part is, to evaluate the influence of the derating setpoint
at different input conditions. The electrical power reduces as expected for all wind speeds. Both, the flapwise and the tower
DELs strongly increase with the wind speed, while the DELs of the edgewise bm reduce when the rated wind speed is reached
at 12 m/s and the turbine starts pitching. The load reduction of the edgewise bm directly corresponds to the reduction in rotor
speed, and thus has a stronger effect at 90 % and 80 % when the rotor speed is lowered by a higher amount than the generator
torque. The decrease of $DEL_{edge}^{st}$ is also rather small compared to the other two failure modes, where the relative difference in
DELs is much higher. The $DEL_{flap}^{st}$ can be reduced for almost all wind speeds. In the partial and the full load region, the load
reducing effect is higher at larger values of $\delta_P$ (80 and 90 %) due to the dependency on the rotational speed. The DELs of the
tower bm show a much less clear relation to the percentage of power. For low wind speeds, the $DEL_{tower}^{st}$ also decrease with





the derating, but with some significant variation within the simulated data points. For higher wind speeds, reducing the power can even increase the tower loads, and the relation is not completely deterministic. As mentioned in Sect. 4.1, this effect is
caused by the reduced aero-dynamic damping due to the rotor speed reduction or from resonance effects. Implementing active damping for the tower or using different control configurations could reduce this effect. It also shows the conflicting behaviour of the different failure modes.

Figure 7 shows the result with the TI on the x-axis for different values of $\delta_P$ with a fix wind speed $v = 8$ %. The power and the $DEL_{edge}^{st}$ are not significantly influenced by the turbulence. The load reduction of the edgewise DEL is low compared to
the other two failure modes. For the flapwise bending moment and the tower bending moment, the strongest relative reduction can be achieved by reducing the power to 90%, but more derating still decreases the DELs slightly further. The relative load reduction also increases with increasing turbulence.

The results presented in this section show several aspects which are relevant for the optimal planning approach. On the one hand, it is possible to determine the relationship between those inputs with a single deterministic function in a simplified way.
Advancements in surrogate modelling, control design and damage modelling can be used to create even more suitable models without changing the overall process described within this paper. On the other hand, the results also show the difficulties which arise from the random turbulent inflow and the conflicts between different failure modes for load reduction. A continuous reduction in power does not necessarily result in a continuous reduction of loads and the high nonlinearity between the inputs, the aero-elastic model and the controller can not fully be covered by a simple regression model. Because the essential part of
the nonlinearity is covered, however, the optimal planning approach can make use of this to determine when a load reduction should be favoured over a higher energy production. This can especially be done by exploiting the fact that higher turbulence significantly increases loads, but the power production remains almost the same. This effect is even strongly enforced from the relation of the short-term DEL to the damage rate because the value is raised to higher power by the Wöhler-exponent (see Eq. (19)). When, and by how much derating is beneficial to apply, can be determined by including the overall frequency
distribution of each situation in the long-term planning approach.

## 4.3 Step 3: Determine optimal condition-based operational strategies for lifetime planning

The general problem, which was formulated in Sect. 2.4, is applied on two different use cases for the example wind farm. At first, the damages of two failure modes are levelled within the example wind farm. As a second step, a Pareto-front is created for each failure mode separately. The results of the Pareto-front are subsequently selected based on an the economic model in
Sect. 4.4.

The optimization problem is solved by using the gradient-based interior point algorithm for constrained non-linear optimization problem (Waechter and Laird, 2022). The problem itself is formulated with Python and the optimizer is interfaced through the library pygmo (Biscani and Izzo, 2020a) which builds on the C++ library pagmo (Biscani and Izzo, 2020b). Gradients are computed by using finite differences.





(a) Power

(b) relative short-term DEL of edgewise bm

(c) relative short-term DEL of flapwise bm

(d) relative short-term DEL of tower bm

**Figure 6.** Evaluated surrogate models (solid curves) and simulated data points (dots) for a fix TI of 11.3%

.

### 4.3.1 Levelled wind farm damage

Before applying the optimization problem, the example wind farm with 9 turbines is investigated with the reference operational strategy $\bar{u}^{ref}$ where all turbines operate at $\bar{\delta}_P = 100\%$ both in partial and in full load operation. The total energy production and the total damages of the considered failure modes are evaluated using Eq. (23) and Eq. (24). The energy production and damages are computed under the assumption that the turbine in the centre determines the design conditions, i.e. the reference $DEL^{ref}$ is computed with the combined wind probability distribution of that turbine including the wake effects. Principally, it means that $\bar{u}^{ref}$ is already adjusted to the specific site. This is clearly not the approach, which is usually applied for design load calculations, but it is advantageous to understand the relative differences between the turbines caused from the wake-induced turbulence. Due to the low assumption for the ambient turbulence, computing $DEL^{ref}$ with the IEC wind class 1A results





(a) Power

(b) relative short-term DEL of edgewise bm

(c) relative short-term DEL of flapwise bm

(d) relative short-term DEL of tower bm

**Figure 7.** Evaluated surrogate models (solid curves) and simulated data points (dots) for a fix wind speed of 8 m/s

.

in significantly lower damage values of all turbines for the tower and the flapwise bm. To show the benefits of the condition

based-optimal planning, the current reference is beneficial, however. Therefore, the total damage of that turbine is equal to 1

for all failure modes in accordance with the introducing explanation in Sect. 2. Consequently, the total damage of each turbine

is always relative to the turbine in the centre with all turbines operating at 100 % because of the relationship determined from

Eq. (19). The resulting damages with all turbines operating with the reference operational strategy $\bar{\delta}_P = 100\%$ in all situations

is referred to as the reference case.

The results for this case are shown in Fig. 8. While the differences in energy production and the damage of of the edgewise

bm lie around 2-4% only (see Fig. 8a and Fig. 8b), there is a strong variation of damage on the tower and the blades in flapwise

direction (see Fig. 8c and Fig. 8d). The high relative differences of more than 0.5 for the flapwise bm and more than 0.3

for the tower bm are mainly due to the wake-induced turbulence. Such high differences consequently mean, that the turbine





with lowest induced damage could be operated much longer than the weakest turbine when only looking at fatigue damage

for instance. Without considering adaptive strategies for the long-term planning, this currently means that all turbines will be

over-designed so that the weakest turbine is able to withstand the induced loads.

(a) Relative energy production

(b) Damage of edgewise bm

(c) Damage of flapwise bm

(d) Damage of tower bottom bm

**Figure 8.** Damage and energy production in the example wind farm relative to the turbine in the centre (turbine no. 4)

.

Through a levelling of fatigue damage of all turbines, a longer operation of a complete wind farm would be possible without

any changes on the structure. For both, the tower and the flapwise bm, the turbine in the lower left corner has the lowest total

damage at about 0.70 of the turbine in the middle. Therefore, the target values for the optimization problem defined in (20) are

defined as $D_{flap}^{target} = D_{tower}^{target} = 0.7$ for each of the 9 turbines except for the lower-left one. The damage of the edgewise bm is





not considered in this case, resulting in the following formulation for the optimization problem:

$$\max_{\bar{u}_s} \sum_{j=1}^{B^v} \sum_{i=1}^{B^{TI}} P(v_j, TI_i, u_s(v_j, TI_i)) \tilde{h}_s^{ref}$$

$$\text{subset to} \sum_{j=1}^{B^v} \sum_{i=1}^{B^{TI}} d_{tower}(v_j, TI_i, u_s(v_j, TI_i)) \tilde{h}_s^{ref} \leq 0.7$$

$$\sum_{j=1}^{B^v} \sum_{i=1}^{B^{TI}} d_{flap}(v_j, TI_i, u_s(v_j, TI_i)) \tilde{h}_s^{ref} \leq 0.7. \tag{32}$$

The results of solving the problem defined in (32) for each of the $s = \{1, \ldots, 9\}$ turbines separately are shown in table 3.

The total damages with respect to the reference case are computed for all of the selected failure modes. For all 9 turbines, the damage can be reduced to the value of 0.7 as specified by the constraints. While the optimization is performed using a probability distribution for the local wind speed and turbulence (from Eq. (25)), the total damage is computed with the distribution for wind speed and wind direction (from Eq. (23)). This causes slight deviations so that the damage sometimes aggregates to a value slightly higher than 0.7, e.g. for turbine number 8. This deviation definitely lies within the margin of

overall uncertainty.

The strong benefits of the approach can be seen when looking at the relative results for each turbine compared to the reference case, as shown in table 4. While the damage of the flapwise bm and of the tower bottom can be reduced at maximum to about 0.54 and 0.70 respectively, there is a much smaller loss in energy production to about 0.93 at maximum. For each turbine, the relative damage reduction of the flapwise bm and the tower bm is higher than the relative energy. Since the loss

in energy only refers to the same time span of $\tau^{ref}$, it would be more than compensated through the extended lifetime from the damage reduction. The lifetime extension factor for a single failure mode is the reciprocal of the relative damage value according to Eq. (9). By levelling the total damage of all turbines in the farm through operational strategies, an alignment of decommissioning or maintenance of the turbines also becomes a major advantage. One also has to state that the damage of the edgewise bm between the different turbines is not aligned anymore. This is due to the strong difference in influence from wind

speed and turbulence intensity on the different loads. Therefore, it is clear that the influence of control on the different failure modes must be carefully balanced and will always depend on the constraints from a specific turbine and site as well as on the specified system boundaries.

To get a better understanding of the results in detail, we exemplarily select the turbine in the center, i.e. turbine number 4. It is the turbine with highest damage on the tower and affected by the wake of turbines from all directions. The distribution of

annual wind frequency, energy production and damage with standard operation are shown in Fig. 9 and Fig.10. In each plot, the wind speed is plotted radially and the wind direction circumferentially. The relative annual frequency at the site is given in percent, i.e. by $100 \cdot p_{\Delta\tau}^{ref}(v^{amb}, \theta^{amb})$. The damage and energy production are given by their total value share on the overall value during $\tau^{life}$, i.e. by

$$d_{fm}\left(f_4^{wake}(v^{amb}, \theta^{amb}), u^{ref}\right) h^{ref}$$





**Table 3.** Result of optimization for each turbine (total damage and energy production with reference from turbine 4 (in the centre))

| Turbine Index | Flapwise bm [-] | Edgewise bm [-] | Tower bottom bm [-] | Total energy [MWh] |
| --- | --- | --- | --- | --- |
| 0 | 0.694 | 1.0 | 0.669 | 900.391 |
| 1 | 0.691 | 0.989 | 0.696 | 879.887 |
| 2 | 0.664 | 0.977 | 0.642 | 878.499 |
| 3 | 0.695 | 0.987 | 0.699 | 874.75 |
| 4 | 0.695 | 0.921 | 0.702 | 808.731 |
| 5 | 0.708 | 0.926 | 0.704 | 815.637 |
| 6 | 0.699 | 0.983 | 0.7 | 876.728 |
| 7 | 0.703 | 0.938 | 0.703 | 825.448 |
| 8 | 0.711 | 0.926 | 0.704 | 821.616 |

**Table 4.** Result of optimization for each turbine (relative damage and energy production compared to operation without derating)

| Turbine Index | Flapwise bm (rel) | Edgewise bm (rel) | Tower bottom bm (rel) | Total energy (rel) |
| --- | --- | --- | --- | --- |
| 0 | 1.0 | 1.0 | 1.0 | 1.0 |
| 1 | 0.838 | 0.991 | 0.885 | 0.995 |
| 2 | 0.644 | 0.974 | 0.824 | 0.982 |
| 3 | 0.859 | 0.987 | 0.836 | 0.992 |
| 4 | 0.695 | 0.921 | 0.702 | 0.936 |
| 5 | 0.58 | 0.921 | 0.719 | 0.934 |
| 6 | 0.79 | 0.981 | 0.833 | 0.987 |
| 7 | 0.652 | 0.937 | 0.718 | 0.949 |
| 8 | 0.537 | 0.919 | 0.721 | 0.932 |


$$P\left(f_4^{wake}(v^{amb}, \theta^{amb}, u^{ref})\right) h^{ref}$$

and respectively.

The highest frequency in the wind distribution occurs in south-western direction (Fig. 9a). This distribution is also strongly reflected in the energy production (Fig. 9b) and the damage of the edgewise bm (Fig. 10a). Contrary to this, the highest amount

of damage for the blades in flapwise direction and the tower is induced in wind direction downstream of surrounding turbines from all directions.

The results of the optimized strategy are shown in Fig. 11. For the plots, the results depending on local wind speed and turbulence are transferred back to values depending on ambient wind speed and wind direction by sorting the results into corresponding bins. While optimization based on local wind speed and direction reduces the number of optimization variables,

an implementation of the strategy based on wind direction is easier to apply in reality in an open-loop setting of the planning approach. Figure 11a shows the optimization variable, i.e. the percentage of power for each input condition. One can clearly





see that derating is mainly applied when the turbine is in the wake of the neighboring turbines. The highest derating of 50% is applied in the combinations of low wind speeds and high turbulence intensities because this is most beneficial for the tower loads. When comparing the optimized damage rose for the flapwise bm (11b) and the tower bm (11c) to the standard operation,

one can clearly see the reduction of damage in the regions with a high damage contribution. Especially when the wind comes from the north or the south, the neighboring turbines are in close distance to the selected turbine and thus damage can strongly be reduced. This optimized derating strategy exploits the fact that reducing damage under those conditions is highly beneficial because a high amount of damage is induced while the energy production is comparably low.

As mentioned before, the derating of each turbine also influences its own wake, which consequently influences the neighbor-

ing turbines. Therefore, the process for levelling the damage of all turbines is not completely fulfilled, and wake effects would need to be recomputed with the new optimized planning strategy or the problem would need to be solved for all turbines at once. It also depends on the selected control setpoints and selected failure modes if the effect of derating on the damage of the turbine itself is higher than the indirect effect through wake reduction on neighbouring turbines. Especially for the tower, this can be highly beneficial following some of the studies mentioned in Sect. 2.2 like Bossanyi and Jorge (2016) or Meng et al.

(2020). Overall, this approach requires an expansion of the system boundaries to a system of interacting systems.

While levelling the damage between all turbines within a specified time period is one use case, another can be determined when looking more closely on a single turbine. Due to the specification of damage reduction, the extended lifetime is actually fixed through the deterministic relationship from Eq. (10). Thus, it is actually neglected as an important degree of freedom. By finding the Pareto-front of energy production and damage, the desired lifetime can be selected on the basis of those results.

The selected turbine in the centre is also used for this second use case.



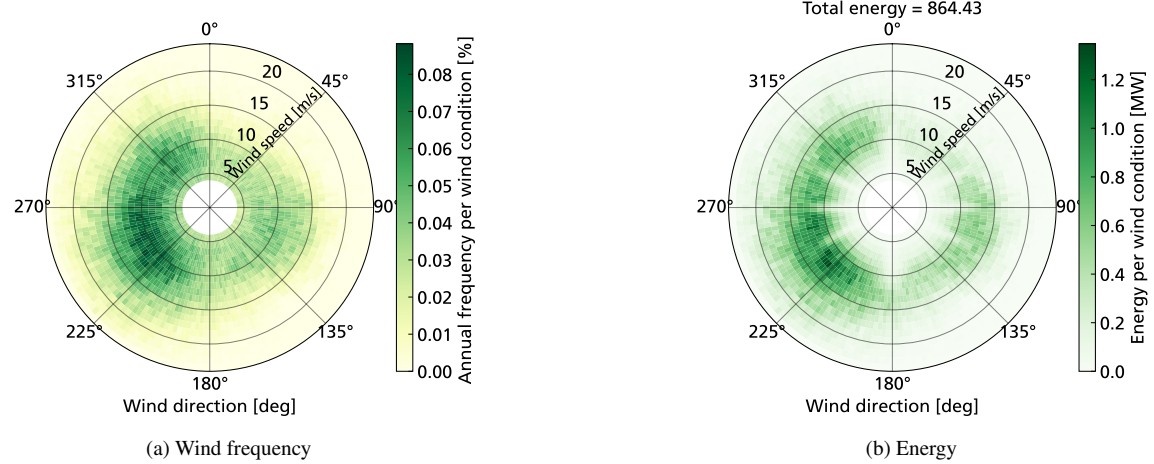

(a) Wind frequency

(b) Energy

**Figure 9.** Distribution of frequency and energy production for all wind directions (plotted circumferentially in degrees) and wind speeds (plotted radially in m/s) for the center turbine of the example wind farm

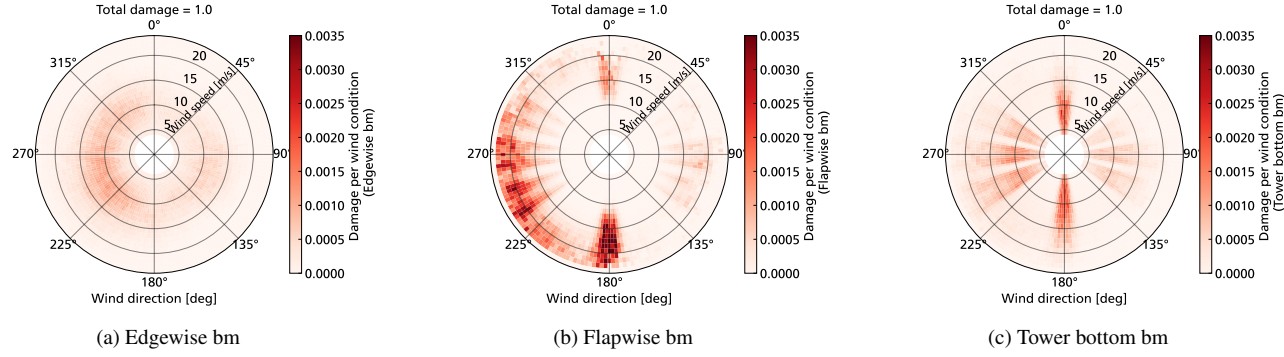

(a) Edgewise bm

(b) Flapwise bm

(c) Tower bottom bm

**Figure 10.** Original distribution of damage (without applying derating) for all wind directions (plotted circumferentially in degrees) and wind speeds (plotted radially in m/s) for the center turbine of the example wind farm

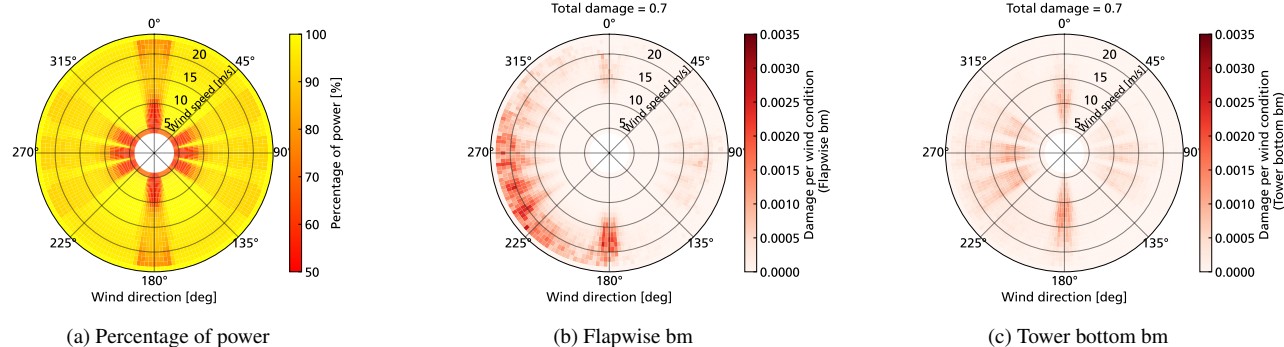

(a) Percentage of power

(b) Flapwise bm

(c) Tower bottom bm

**Figure 11.** Optimized derating strategy and associated distribution of damage for all wind directions (plotted circumferentially in degrees) and wind speeds (plotted radially in m/s) for the center turbine of the example wind farm



### 4.3.2 Pareto front for energy and damage

As it was mentioned in Sect. 2.4, the Epsilon-contraint method can be directly applied on the optimization problem defined in (20) to find a Pareto-front. By solving the problem for various values of $D_{fm}^{target} \in [0,1]$, the maximum amount of energy for each of these values can be found. For simplification, each failure mode is considered separately. On the one hand, this increases the interpretability of the results. On the other hand, it would be applicable if the weakest failure mode of a turbine or component can clearly be determined.

For each failure mode $fm \in \{flap, edge, tower\}$, at first the minimum possible damage is computed as an orientation. Then the optimization problem

$$\max_{\hat{u}} \sum_{j=1}^{B^v} \sum_{i=1}^{B^{TI}} P(v_j, TI_i, u(v_j, TI_i)) \tilde{h}_4^{ref}(v_j, TI_i)$$

subset to $$\sum_{j=1}^{B^v} \sum_{i=1}^{B^{TI}} d_{fm}(v_j, TI_i, u(v_j, TI_i)) \tilde{h}_4^{ref}(v_j, TI_i) \leq D_{fm}^{target}. \tag{33}$$

is solved with $D_{fm}^{target} \in \{0.65, 0.7, \ldots, 1\}$ for the $fm \in \{tower, edge\}$. For the flapwise bm, the damage can be reduced down to a minimum of 0.24, so that the values $\{0.25, 0.3, 0.4, 0.5, 0.6\}$ are added to the previous set to define the values of $D_{flap}^{target}$. Therefore, several optimal planning strategies are computed separately for each failure mode denoted as $\bar{u}_{fm}^{opt}$.

The results of the optimization for each failure mode are shown in Fig. 12 where the relative energy production is plotted over the relative damage. When comparing the results, one can clearly see the different behaviour of the failure modes, which results from the determined relation of the damage rates to the control setpoints and the external conditions. While it is possible to significantly decrease the damage of the flapwise bm (Fig. 12c) and the tower bm (Fig. 12b) without losing much energy, the edgewise damage can only be reduced with comparable losses in the energy production. This is mainly due to the fact, that the dependency of the edgewise bm on TI is low and that damage can mainly be reduced by reducing the rotational speed, as it was partially discussed in 4.2. The strong relation of energy production and induced damage of the edgewise bm can also well be seen be comparing Fig. 9b and Fig. 10a. Therefore, the applied method has a much bigger potential for turbulence induced loads, as it was already expected from the behavior for each of the DELs, as presented in Sect. 4.2.

As mentioned earlier, reducing the damage results in a factor for lifetime extension which is approximately determined by Eq. (9) under the limitations discussed in Sect. 2. According to Eq. (11), the energy yield after the extended lifetime $\tau_{fm}^{life}(\bar{u}_{fm}^{opt})$ is also increased by that factor. Additionally, the selected failure mode is assumed to be the only one relevant to life extension so that the damage of the others can be neglected for this example. By directly maximizing the energy production, the maximum amount of energy can be produced while fully using up the fatigue budget of the failure mode with a variable time span in this case. The result of this optimization is shown as an orange dot in Fig. 12 and Fig. 13. Figure 13 additionally shows the relative energy production for each failure mode plotted over the relative damage.

For the edgewise bm, only a slight increase of the energy production of about 2% can be obtained when the damage is reduced between 0.85 and 0.9. From below 0.75, the damage reduction even results in a loss of energy and is thus not effective anymore. For the tower bm and the flapwise bm, the amount of damage reduction is always stronger than the loss in energy





so that the overall energy production after the extended lifetime can be significantly increased. For the tower, reducing the damage down to 0.68 results in the highest energy gain of 35%. A further damage reduction reduces the effect slightly. The

strongest positive effect can be seen on the flapwise bm due to the combined influence of the selected control method, the strong influence of high wind speeds and turbulence, as well as the high Wöhler-exponent. The damage can be reduced down to a value of 0.25 resulting in an increase of energy by more than factor 3. While the additional energy production for the tower bm almost increases linear at first and then reaches the maximum value at 0.7, it clearly shows a more than linear growth for the flapwise bm. However, the loss of energy and thus the extension factor have a similar value for both, the flapwise and the

tower bm When the damage is reduced down to 0.7.

Overall, the results show that effectively reducing the damage has a large potential to increase the energy production from a given budget. This especially holds true for certain failure modes where derating is more effective under high loading external conditions. The computed Pareto-fronts represent a trade-off between damage and energy production over a given time period. Selecting which trade-off is the best can additionally be assessed by an economic evaluation, which is conducted in the

following section.

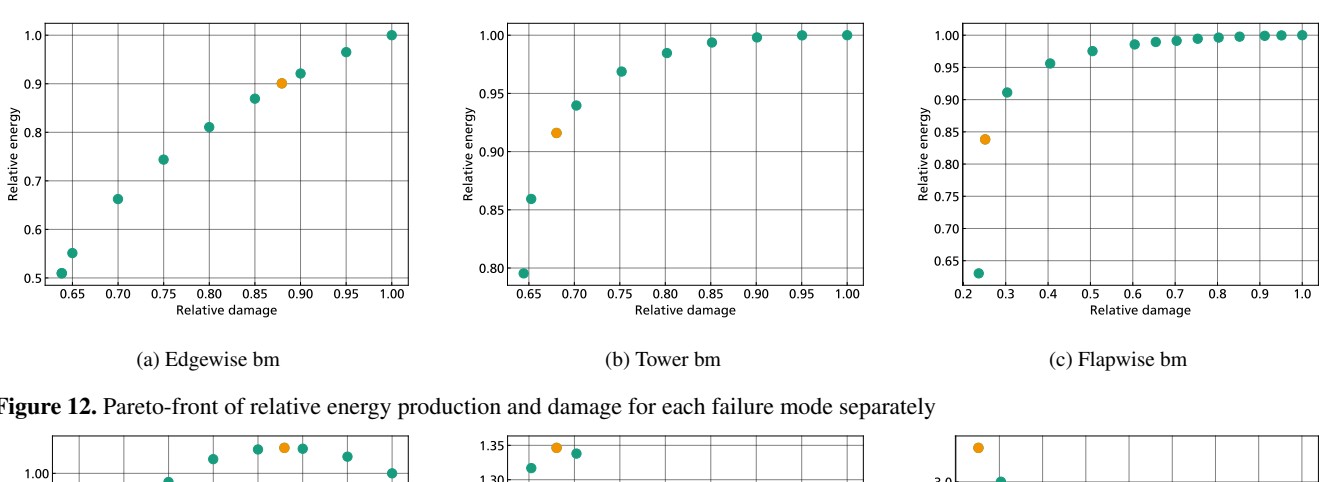

(a) Edgewise bm       (b) Tower bm       (c) Flapwise bm

**Figure 12.** Pareto-front of relative energy production and damage for each failure mode separately

(a) Edgewise bm       (b) Tower bm       (c) Flapwise bm

**Figure 13.** Relative energy over damage plotted over the relative damage for each failure mode separately



## 4.4 Step 4: Select economically best operational lifetime-planning strategy

In Sect. 4.3.2, each of the three failure modes were already considered separately. An economic evaluation is most important for tower damage. A tower replacement is usually considered to be infeasible, which in turn determines the possible lifetime of the entire wind turbine. An exchange of the rotor blade, in contrast to this, can be a feasible approach to extend the turbine's
lifetime when one of its failure modes has reached its fatigue budget. Having this in mind, it is still advantageous to create a planning for these replaceable components in order to coordinate the replacement of several blades or to find the best timing. Considering all of these aspects would require further detailed models on component costs and the specific situation of a wind farm. A second reason, why the tower damage is most suitable for the economic evaluation, can be derived from the previous results. Due to high damage reduction of the flapwise bending moment, very long lifetime extensions would be possible. In
contrast, the edgewise bending moment only offers a small potential for a beneficial lifetime extension. The major drawback for the consideration of the tower loads is the fact, that the influence of the selected derating method on the tower loads is subject to the highest uncertainty among the three failure modes and can sometimes even have negative impact, as it can be seen from the results in Sect. 4.2. Despite that, the optimal planning results from Sect. 4.3 show that it is still possible to significantly reduce the damage when it is applied under suitable conditions. Reducing the uncertainties would be required for an adaption
to a real turbine.

In order to select a trade-off based on the economic benefits, the net present value from Sect. 2.5 is used. The average costs for a wind farm are taken from BVG Associates (2019); they are summarized in Table 5. All values are scaled to a single turbine with 7.5MW power. The numbers are actually valid for a full wind farm, so that scaling it to a single turbine is not fully realistic. It can rather be seen as an examination of the entire wind farm "per turbine". Therefore, all of these values are very
rough assumptions which just allow for the possibility to compute the potential increase in profit within a realistic range.

**Table 5.** Overview of parameters for financing model

| CAPEX per MW | OPEX | Change rate | WACC |
| --- | --- | --- | --- |
| 2.37 Mio £/MW ≈ 2.73 Mio. €/MW | 76 k£/MW ≈ 87.4€/MW | 1.15 €/£. | 6 % |

The annual income is computed with the reference annual frequency distribution and the operational strategies from the results. An availability factor of 0.95 is assumed. In addition, we assume an electricity price of 0.066 €/kWh at which the wind farm is barely able to recover the investment cost after a lifetime of 25 years, when being operated with the reference strategy $\bar{u}^{ref}$. The financing model using the NPV from Eq. (21) is applied to all of the derating strategies which were computed for the
tower in Sect. 4.3.2. The lifetime of the turbine is always determined as the time after which the induced damage has reached the fatigue budget, i.e., by Eq. (10). For the final year, the annual income is computed as a fractional value, depending on relative damage increment before the value of 1 is reached. Here, the seasonal variations discussed in Sect 2 are neglected.

The results are shown in Fig. 14. In all three subfigures, the green curves correspond to the Pareto-optimal points from Fig. 12b. The orange curves highlight the one trade-off, where the maximum energy is being produced over the extended
lifetime, which is almost 37 years. The red curves highlight the operational strategy with best economic results, i.e. the highest





NPV at the lifetime where the damage equals 1. It results in a relative damage value of 0.8 and an extended lifetime of slightly more than 31 years. Since the same frequency distribution for wind conditions and the same operating strategy is assumed for each year, also the annual damage and annual energy production are equal. This results in a linear increase of the damage in Fig. 14a and the energy production in Fig. 14b. Fig. 14c shows the net present value representing the permissible investment if the system was operated until a certain year.

The assumed initial costs (CAPEX) are equal to about $7.5\ MW \cdot 2.73$ €/MW $\approx 20.4\ M$€. It can be seen clearly that for maximum energy generation, the system has to be operated at least 35 years until it is economically viable, whereas the reference strategy and the economically optimal strategy require only about 25 or 26 years of operation respectively. At 35 years, the NPV of the strategy which maximizes energy is nevertheless about 0.25 M€ higher than the value with the reference strategy at 25 years. The difference of the strategy which maximizes NPV at a lifetime of 31 years to the reference strategy at 25 years is about 1.39 M€.

Therefore, both optimized strategies will pay off after a longer operating time, but the strategy maximizing NPV leads to a significantly higher NPV at an earlier time. This can also clearly be seen from Fig. 14c. The reduced energy yield per year of strategy maximizing total energy (orange curve) leads to lower income, lower repayment per year and in turn to lower NPV over the entire lifetime compared to the strategy which maximizes NPV (red curve),

For all strategies, it must be noted that the assumed WACC of 6% needs to be taken into account as well. while the net present value does not change significantly in later years, the profit would increase strongly once the investment has been repaid. Thus, the actual profit can be multiplied with $(1,06)^r$ where $r$ is the number of years operating once the investment hast been returned. Therefore, a slightly higher NPV can already result in much higher profits.

Overall, the assessment of economic benefits always needs to be done under consideration of the specific assumptions and parameters for a specific project and can be done in much more detail. Especially the price of electricity underlies a high uncertainty and can hardly be predicted for 30 years in the future. Nevertheless, the exemplary evaluation shows how multiple optimized planning strategies can be used to obtain an economically optimized solution, depending on the objectives and input parameters.

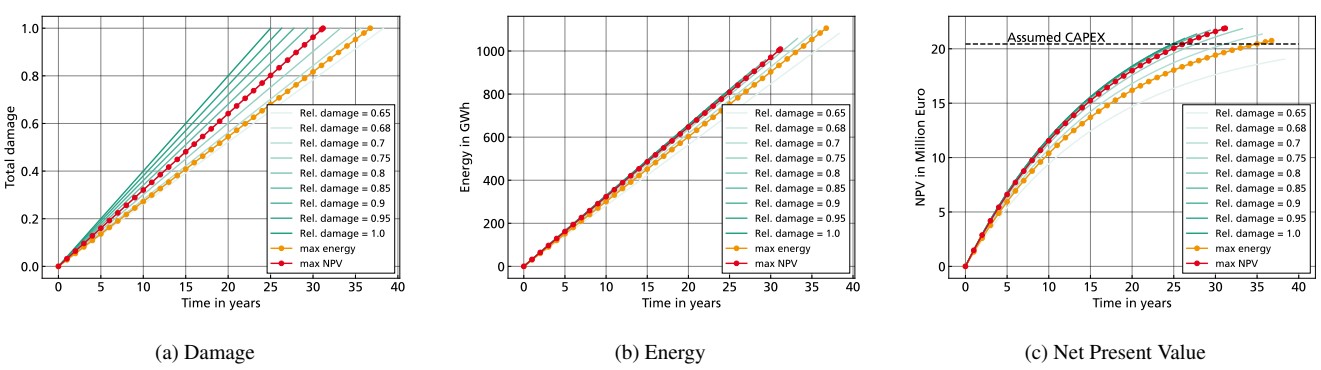

|(a) Damage|(b) Energy|(c) Net Present Value|

**Figure 14.** Annual progression over time for damage energy and profit for multiple optimized planning strategies (Green: results of Pareto-front; Orange with dots: Maximum energy production; Red with dots: Maximum profit)



# 5 Discussion


With the application example, we aimed at showing that all four identified steps build on one another and how the process can be used for an effective distribution of damage over the entire lifetime of a turbine. This way the interaction of the inputs, such as control setpoints, environmental conditions, damage progression and energy production, becomes clear. Step 1 establishes the connection from selected setpoints on supervisory control level to the effect these have on the real-time controller. The

transfer of the setpoints to parameters of the real-time controller results in changed loads and power production. This leads to a change in the behavior of the damage rates depending on the external conditions, which is expressed by the surrogate models created in step 2. Therefore, the selection of a control setpoint determines how effective the damage can be reduced for a single input condition at a time scale of $\Delta t = 1h$. The distribution of the damage progression follows for the long-term perspective by determining the optimal strategies in step 3. This step was conducted for two different use cases. The first use case, where the

damage of all turbines within the farm was reduced to the same level, is not followed by the fourth step because the technical objective is, for the time being, independent of the economic evaluation. The main purpose of this use case was to show the technical benefits of the optimization approach. For the second use case, where multiple planning strategies are optimized for a single failure mode of one turbine, adding the fourth step allows for an economic evaluation of the strategies at first and subsequently helps to select the best overall trade-off.

Within the specified system boundaries, the advantages of the approach become evident in both use cases. As a result of the condition-based optimization, the induced damage is saved exactly where the relation to the energy production is unprofitable. This can explicitly be observed on the results of a single turbine, as shown in Fig. 11. While the turbine is operated with its reference control at $\delta_P = 100\%$ most of the time, it is not worth inducing a high amount of damage when the energy production is comparably low. This is mainly done when the turbine operates in the wake of another turbine. Principally, the result could

be interpreted as a sector management plan. In contrast to simple sector management schemes, the planning is provided in such way, that the amount of damage reduction for each inflow condition is just sufficient to meet the overall damage target while energy production is maximized.

In each of the optimizations carried out, either for levelling the overall farm damage or to create a Pareto-front of multiple results, the individual optimal distribution for the specified objective and constraints is found. The importance of a suitable

specification of these becomes apparent with the second use case because multiple trade-offs between damage and energy production can be found for each failure mode. This use case also shows that it is not only possible to gain more energy from the given fatigue budget, but also to create a significantly higher profit. Both, the increase of energy and profit, result from lifetime extension due to overall damage reduction as it can be visibly observed in Fig. 14b.

Through the introduction and the description of the theoretical background of the individual steps, it has also become clear

that the results are generally subject to some assumptions and that some influencing factors are neglected. Furthermore, the results of the example always depend on the defined system boundaries, which additionally do not take some influences into account. In principle, therefore, a distinction must be made between limitations and potential improvements of the approach itself and limitations for the specific application example.





The approach relies on the use of surrogate models, which have a limited validity and implement a deterministic relationship
from inputs to damage rates. Therefore, the results are only valid for the utilized model. The use of various surrogate models
for the computation of site-specific damage has extensively been discussed in Sect. 2.3. Without the use of surrogates which
explicitly include the influence of the turbine controller, only a single operational strategy can be employed. The selection
of a single reference operating strategy for the turbine also relies on assumptions about future wind conditions and damage
progression, e.g., specified by standard IEC design wind classes. Thus, by involving adaptive control as an instrument, a site-
specific operation of a turbine under specified targets becomes possible, and surrogate models are suitable for its optimization.
In the end, the uncertainties of the surrogate model for the adaptive planning need to be estimated together with other design
and operating assumptions. Those need to be considered for the specification of suitable safety margins, together with the
possibilities of re-adapting the planning based on the long-term performance evaluation of the turbine, changed long-term
targets or improved surrogate models.

Besides this general dependency of the approach on the surrogate model, there are some limitations in each of the four steps,
which influence the results. When the setpoints for the real-time controller are provided in step one, the degrees of freedom for
the optimization is limited by the choice of a single setpoint. On the one hand, the applied derating strategy can be fine-tuned
to create a higher load reduction on the tower, for example. Also, a power-boost could be included as a percentage factor. On
the other hand, adding additional control setpoints increases the possibilities of planning the damage progression, but also the
number of inputs for the surrogate model and in turn the number of optimization variables.

The choice of input conditions is part of the specification of system boundaries and has an influence on the results of the
use case. Within the application example, the system boundaries have been defined quite narrowly to explicitly demonstrate
the approach for wake induced damage increases. Adding further external input conditions is explicitly covered by the general
approach. This is mainly done in step 2, because each input requires an extension of the surrogate model for this degree of
freedom and the underlying simulation model needs to be able to cover the effects of those inputs. Covering the wake effects
through wake induced turbulence is a strong simplification which neglects the dynamics of the wake and partial wake effects.
Also, wind shear and yaw-misalignment as well as grid events and idling could also be included. For offshore wind turbines,
including a model of the ground, the foundation in combination with wave influence would be required because reducing
the aero-dynamic damping through control can have an even higher negative influences than it was observed in the present
example. Overall, the determination of the relevant input conditions for the system boundaries must always be considered in
conjunction with the possible setpoints for the real-time controller, since it is this coupling that is ultimately exploited for
optimized planning.

In addition to the consideration of external inputs in the surrogate models, including them within the statistical frequency
distribution of the condition is also required. Here, one needs to distinguish between conditions which are influenceable by the
setpoints of the controller during power production, such as the additional wind influences mentioned above, and conditions
that cannot be influenced (e.g., idling), but which nevertheless participate in the overall damage process. For the application
example, including a frequency distribution of the ambient turbulence would increase the accuracy of the results, as a first





step. For the choice of inputs, one needs to balance between accuracy, computational cost and relevance of each input for the damage progression and the setpoints of the real-time controller.

Furthermore, setting the system boundary at a single turbine is a limitation that has already been mentioned. Solving the optimization problem for each turbine individually neglects the influence of the turbine controller on the wake and thus on the surrounding turbines. To include this effect, the system boundary needs to be extended to a complete wind farm and the optimization problem would need to be formulated for all turbines at once. Then, the control setpoint of the upstream turbine influences the local input conditions and thus the damage progression of the downstream turbine. This way, it would also be

possible to combine derating with wake steering for damage reduction and power maximization. However, this approach has the drawback that the number of optimization variables increases. At first, the local frequency distribution created in Sect. 3.2.3 cannot be used anymore to decrease the number of optimization variables by binning the wind conditions into wind speed and TI because the wind directions need to be considered within the optimization problem. Additionally, the number of variables is proportional to the number of turbines. Overall, solving the problem for a complete wind farm including the interactions

through control for all input conditions at once is a feasible approach, when sufficient computational power for the increased number of optimization variables and input conditions is available. Using an iterative approach where the resulting planned operational strategies for each turbine is fed back to the wake modelling program would be possible.

  Within step 3, i.e. the determination of the condition-based strategy through mathematical optimization, the formulation of the target damage as constraints is a limitation of the approach because it requires a preselection of the target damage values.

This might not always lead to optimal or even feasible solutions. Specifying the target values for an increasing number of failure modes resulting from wider system boundaries, e.g. by considering a system of systems like a wind farm, thus s requires a high understanding of the system and the pursued targets. Levelling the damage of one or more failure modes of all turbines is still a valid approach, but might not be achievable in all cases. Therefore, selecting a solution based on multiple optimized strategies under consideration of further information like economic factors can overcome this issue. Within this work, multiple strategies

constrained by different damage targets were found to create a Pareto-front in Sect. 4.3.2. This approach is currently restricted by using a single failure mode.

  A combination of step 3 and 4 might be a potential approach to create an integrated solution which includes finding suitable damage targets for each failure modes. This way, component costs could also be considered. However, combining these steps hast two major drawbacks. At first, it increases the complexity of optimization problem, and it requires the consideration of time, because one component can fail earlier than another. Then, the possibility of replacement or repair would need to

of time, because one component can fail earlier than another. Then, the possibility of replacement or repair would need to be integrated. Since replacement will lead to a jump in maintenance costs, it makes optimization more difficult. In addition, considering the probabilistic nature of failure would further increase complexity. Resolving the separation of the technical from the economic level can be seen as a second drawback, because it can reduce the interpretabiliy of results. On the other hand, still separating the technical and economic part for a large number of turbines and failure modes would require to create

a many-objective Pareto-front and does not necessarily increase interpretabiliy. Solving this issue is still part of upcoming research.



Within the application example of this work, selecting the trade-off based on a simple repayment model using the net present value in step four only covers a small of the economic aspect. It does not cover any variable costs for maintenance or prices of electricity. With this approach, it is possible to show that a reduction of the energy production in some situations can still pay off through the extended lifetime. Finding an integrated solution for the economic benefits, in combination with the target values for the damage and the component costs, still requires advancements of the approach. A first extension of our current approach is possible, by allowing an annual selection of the trade-off between energy and damage on the Pareto-fronts, which are computed in Sect. 4.3.2. This way, it would be possible to allow a higher damage progression at the beginning to reduce the interest burden and to reduce the damage progression of the turbine later on.

Regardless of the limitations, the results of the application example show how a condition-based approach for long-term planning can be used to achieve a targeted fatigue damage planning. It balances the trade-off between induced damage and energy production under the given system boundaries and constraints of the application example optimally.

It is possible to apply the method to a real-world scenario when the system boundaries are well-defined and adapted to the specific use case. In a first step, the provided planning strategy can be used in an open-loop scenario, where the turbine follows the setpoints of the planning. This requires a sufficiently accurate measurement of the input conditions which are used for the planning. The bin width of the optimization could be adapted according to the measurement accuracy. The open-loop approach can be used, to extend the lifetime after the turbine has already been operating for a significant time span. If the approach is applied during the design process, it could be used to save material through a less conservative design. Therefore, the planning approach already brings a high potential even without a closed-loop operation in combination with reliability-control. For applying the approach, some mentioned uncertainties would need to be reduced by adapting the specific use case. Also estimating the amount of uncertainties would be required.

Relating the approach back to the context of reliability-adaptive control, the damage contribution in each wind condition can be used as a setpoint for the closed-loop controller, which is presented on the right-hand side of Fig. 1 in Sect. 1.2. In this case, the planning would not provide a setpoint for the controller itself, but the control loop on the operating stage would try to stay as close to the provided planning based on the performance evaluation of the wind turbine or wind farm. The evaluation would need to include the health-status of the component. In a first step, this could be covered by using the same surrogate models as for the planning, but with the actual wind conditions as an input. Overall, the open-loop approach using the optimized operational planning already creates significant benefits, which can be further increased by using it for a closed-loop operation. The challenges to do so were partially discussed in Sect. 1.1.

# 6 Conclusion and Outlook

We presented a novel approach for an optimal planning for the operation of wind energy systems over their entire lifetime. This was achieved by introducing a four-step process, of which the key is to formulate a mathematical optimization problem which optimally distributes the available damage budget of a given failure mode over the total turbine lifetime. Within the





introduction, the objectives for this work were derived from the context of reliability(-adaptive) control. A planning, which
pursues long-term objectives of operation, was identified as an important prerequisite.

Following this, the four steps to provide an optimal planning were introduced, starting from providing setpoints for the
real-time controller of a wind turbine (step 1), continued by their usage as an input for the creation of surrogate models for
the induced damage (step 2). Subsequently, those surrogates are used to determine optimally planned operational strategies as
results of a nonlinear optimization problem (step 3). The selection of the results is based on an economic evaluation as a final
step of the approach (step 4). The theoretical background for each of the steps was provided, including literature researches;
then, the method was applied to an example for demonstration. Within this work, the process is focused on the planning of the
fatigue damage progression of different wind turbine failure modes. The application example serves as a proof-of-concept for
the process. It shows the high potential of the approach for an effective damage reduction for two different use cases.

The optimization approach in step 3 allows to meet the specified targets of the use cases, because the damage budget can
be saved or spent depending on operation conditions such that it pays off most in the long-term perspective. This way, it is
possible to gain more energy from a given system and thus to reduce cost and ecological impact by a better usage of materials.
Overall, the long-term planning brings great-advantages in itself and still offers a high potential for further development of the
approach.

Limitations to the approach were discussed in great detail in Sect. 5. These also provide starting points for potential improve-
ments. In general, the separation into four steps allows to create intermediate solutions and to interpret each of them. But it also
creates some limitations in itself, like the restriction to pre-selected setpoints for the real-time controller. A higher integration of
the steps, e.g., building or improving surrogate models within the optimization process, could be possible, but requires higher
computational power and leads to reduced explainability. The current process thus represents an approach which can be used
with currently available methods, despite the mentioned drawbacks. It is applicable with reasonable computational power, and
each of the steps can be verified individually. Also, the four steps could principally be applied to most existing turbines with
minor adjustments.

The main field for improvement of the four-step method is to allow it to operate within a broader system boundary. At first,
this means that an extension to optimizing for multiple failure modes at once is required. Secondly, optimizing for coupled
systems, e.g., wind turbines that interact through wind farm control, must be possible. In addition, a more advanced financing
model might reduce uncertainty and thus yield further benefits for operations. Ideally, this could also include fluctuating
electricity prices, which we did not yet take into account.

To apply the strategy, a coupling of the planning stage with the operational stage is required. As a first step, an open-loop
implementation can be implemented. To do so, the properties of a specific wind turbine or wind farm need to be identified
and coupled with the planning approach. An adaption of the planning after some time, as also indicated in Fig. 1 with the
arrow for "Readjust planning", would allow for a simplified "continuous" adaption of the system based on the current system
performance.

In such a scenario, it needs to be examined how short-term deviations from the planning, e.g., by reacting on electricity
prices or simply on grid requirements, can be tolerated while at the same time following the provided planning sufficiently



well. The best time and way to readjust the planning also needs to be investigated. Connecting the operational planning with additional inputs like maintenance planning would bring further advantageous to the approach. A real-closed loop behavior, where the planning provides setpoints for a reliability controller, has an even higher overall potential but also brings further challenges which were discussed in the introduction already (Sect. 1.1).

One aspect, which can not be covered by the current planning approach, is considering sequence effects, i.e., dependencies of future damage progression on previous damage. By binning the relevant deterministic conditions and considering their frequency distribution, the linearity of damage progression is always assumed for long-term effects. In this case, the approach can at least be used as an initial planning step, partially covering the linear part of a damage progression process.

In addition, the probabilistic nature of failure will need to be addressed for further advancements of the approach, e.g., through sufficient safety margins or by adding uncertainties to the optimization. This is not only caused by the probabilistic nature of failure, but also because the uncertainties of wind prediction and electricity prices cannot be eliminated completely.

In the future, the coupled operation of wind turbines or wind farms with power-to-X systems will become highly relevant. This increases the need for adaptive operation because the damage progression of connected systems also needs to be considered and the question when to operate each system on what level needs to be answered. Therefore, such a coupled operation leads to a further expansion of the system boundaries and brings more complexity on different levels. For hydrogen production, the damage progression in an electrolyzer needs to be integrated in order to assess their reliability. It is also necessary to include prices for selling hydrogen and thus serve a second market. As a first step, it needs to be examined in what way a coupled operation of a wind turbine with an electrolyzer could be influenced by different setpoints of the real-time controller and how this can be used to influence the damage progression of both systems.

Concluding, the presented work provides an applicable and adaptable method for wind turbine operation. It still offers possibilities for further improvements on various levels, as well as potential for research and further development in different areas. More research is needed to reduce uncertainty and consider multiple components and failure modes in the approach on the planning level. Additionally, the integration with reliability-adaptive control offers further advancements to discover the full benefits for a more sustainable wind farm operation.



## List of symbols

| | |
|---|---|
| $fm$ | Failure mode |
| $\bar{u}^{ref}$ | reference operational strategy |
| $\tau^{ref}$ | reference lifetime |
| $\bar{u}^{opt}$ | optimized operational strategy |
| $\bar{u}$ | arbitrary operational strategy |
| $\tau^{life}$ | free modified lifetime |
| $\Delta\tau$ | time increment for long-term planning, e.g. 1 year |
| $Y$ | number of time increments |
| $c^{ext}$ | extension factor |
| $c_{fm}^{ext}(\bar{u})$ | extension factor for failure mode $fm$ depending on operational strategy |
| $\tau_{fm}^{life}(\bar{u})$ | chosen operation period for failure mode $fm$ depending on operational strategy |
| $\Delta t$ | time increment for the definition of input condition, e.g. 1 hour |
| $x \in X$ | external input conditions valid for a time of $\Delta t$ |
| $\bar{x} = \{x_j\}_{j=1}^{B^x}$ | set of input conditions with $B^x$ bins |
| $B^x$ | number of bins for all input conditions |
| $x_j$ | input conditions for bin $j$ |
| $w = Dim(X)$ | number of independent wind conditions |
| $B^{x^{(w)}}$ | Number of bins defined for condition $x^{(i)}$ |
| $u(x) \in U$ | setpoints for real-time controller depending on x |
| $\bar{u} = \{u(x_j)\}_{j=1}^{B^x}$ | definition for operational strategy as set of setpoints depending on $\bar{x}$ |
| $p_{\Delta\tau}(x)$ | relative frequency distribution of input conditions |
| $h_\tau(x)$ | absolute frequency distribution of input conditions for a time period $\tau$ |
| $h^{ref}$ | reference absolute frequency distribution which is applied for planning of a site |
| $D_{fm}(\tau^{ref}; \bar{u}, h_\tau)$ | function for damage for a failure mode with variable $\tau$ |
| | depending on the operating strategy and the frequency distribution as parameters |
| $\Delta D_{fm}(\bar{u}, h_{\Delta\tau})$ | damage increment for strategy $\bar{u}$ (time increment $\Delta\tau$) |
| $d_{fm}(x, u)$ | damage rate for failure mode $fm$ (time increment $\Delta t$) |
| $P(x, u)$ | power production under the input conditions (time increment $\Delta t$) |
| $E(\tau; \bar{u}, h_\tau)$ | function for energy production with variable $\tau$ |
| | depending on the operating strategy and the frequency distribution as parameters |
| $\Delta E(\bar{u}, h_{\Delta\tau})$ | damage increment for strategy $\bar{u}$ (time increment $\Delta\tau$) |
| $n_i$ | Number of load cycles |
| $N_i$ | Maximum bearable number of load cycles |





| | |
|---|---|
| $D^{ult}$ | ultimate design load |
| $m$ | Wöhler coefficient |
| $L_{ij}$ | Oscillation amplitude of a load cycle |
| $N^{eq}$ | Number of equivalent load cycles |
| $DEL^{st}(x,u)$ | short term damage equivalent load |
| $DEL(\tau,\bar{u})$ | lifetime damage equivalent load |
| $DEL^{ref}$ | reference damage equivalent load |
| $z = (x, u(x))$ | input to surrogate model as combination of external conditions and control setpoints |
| $\hat{z} = (\hat{x}, \hat{u}(x))$ | input sampling for the creation of surrogate models |
| $f_{DEL}(z)$ | surrogate function for $DEL$ of failure mode $fm$ |
| $f_P(z)$ | surrogate function for power production |
| $D_{fm}^{target}$ | target fatigue budget for failure mode $fm$ |
| $NPV(Y)$ | Net Present Value depending on year $Y$ |
| $C_{elPrice}$ | constant electricity price |
| $C_{OPEX}$ | constant annual costs for operation and maintenance |
| $c_{WACC}$ | constant interest rate defined as weighted average costs of capital (WACC) |
| $v^{amb}$ | ambient wind speed |
| $v$ | local wind speed |
| $TI$ | local turbulence intensity |
| $\theta^{amb}$ | ambient wind direction |
| $s$ | turbine index in a wind farm |
| $S$ | number of turbines in a wind farm |
| $f_s^{wake}(v,\theta)$ | wake calculation function for a turbine $s$ |
| $B^{v^{amb}}$ | number of bins for ambient wind speed |
| $B^{\theta^{amb}}$ | number of bins for ambient wind direction |
| $B^v$ | number of bins for local wind speed |
| $B^{TI}$ | number of bins for local turbulence intensity |
| $M$ | generator torque |
| $k$ | generator torque coefficient |
| $M_r$ | rated generator torque |
| $\omega_r$ | rated generator speed |
| $\delta_P$ | percentage power factor |
| $\beta$ | pitch angle |
| $\lambda$ | tip speed ratio |
| $P_r$ | rated power |
| $\delta_\omega$ | percentage generator speed factor |
| $q$ | vector of coefficients of the polynomial for a surrogate model |



*Code and data availability.* Codes and data were created for each steps of the published method. However, some of the tools used are very specific to an in-house workflow and are only partially documented. Thus, there is no single self-contained program of which sharing would provide value for others. The dataset of load simulations will be shared publicly with the final revised paper. For inquiries about data or code, please reach out to the authors.

*Author contributions.* NR and TM developed the concept for the study and the research methods. NR developed the code and produced the results under supervision and with valuable input from TM. TM was responsible for project administration and funding acquisition. NR wrote the draft version of the paper. RH reviewed the draft version and gave valuable feedback on the draft, the concept and the results. TM and NR reviewed and edited the draft version.

*Competing interests.* The authors declare that they have no conflict of interest.

*Acknowledgements.* The research was carried out by Fraunhofer IWES under the framework of two research projects funded by the Bundesministerium für Bildung und Forschung (BMBF): *H2-DIGITAL* (grant no. 03SF0635) and *Verbundvorhaben H2Mare_VB0: OffgridWind* (grant no. 03HY300E). Additionally, parts of the work were conducted within the project *DigiWind* in cooperation with TU Wien. TU Wien was funded by VGB Powertech e.V. in *DigiWind*.





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
