# Peer review of "From wind conditions to operational strategy: Optimal planning of wind turbine damage progression over its lifetime"

_Wind Energy Science, 2022_

## Referee Comment (RC1)

**Review of "From wind conditions to operational strategy: Optimal planning of wind turbine damage", manuscript id: wes-2022-99**

Reviewer: Vasilis Pettas, University of Stuttgart

**Summary**

The article suggests and evaluates a novel operational method for wind turbines within a wind farm. The long-term fatigue consumption and energy production of each turbine is managed individually by adapting the power output level according to wind conditions to reach lifetime objectives. The motivation for such an adaptation is explained along with all the steps of the process including the controller re-design, the surrogate modeling and the relevant optimization techniques. The results are demonstrated with example use cases, showing the potential applications for equalizing the fatigue consumption among the turbines and a Pareto analysis on fatigue consumption reductions which are then converted to lifetime extension potential and eventually translated to potential economic benefits.

**General Comments**

This is an interesting article with a novel approach to wind turbine operation showing an alternative to the single-dimensional current operational paradigm. The authors have put a lot of thought and effort into the manuscript addressing a multidisciplinary topic. It has a thorough literature review, the research question is framed sufficiently, the methods used are explained adequately, the language used in terms of syntax and typos is in general good and the topic is relevant to the community and the scope of the journal.

I have concerns about some of the assumptions used, especially for the economical evaluation of lifetime extension, but I reckon that this is a proof of concept and I suggest that these are discussed rather than directly addressed. My major concern is about the readability and clarity of the manuscript. The article is too long with a lot of repetitions and overlaps between the sections making it cumbersome to understand and difficult to navigate through. Moreover, the authors explain extensively methods that are not in the scope of the article or give much more detail than required in places. I think the article has to be restructured, sharpened and significantly decreased in length to be suitable for publication as a journal article.

I believe the manuscript is relevant and should be published after addressing the issues mentioned here through a major revision. Find more specific comments below.

**Specific comments**

1. Sections 1.3, 2 and 4 have a lot of overlaps, repetitions and some general discussions being more than 26 pages combined. I suggest merging them and discussing each topic only once. E.g.: Show the general method briefly (similar to section 1.3) as an introduction and merge there sections 2, 4.1 and 4.2. Then make a new section only with the results of the use cases (sections 4.3, 4.4). This is an example of possible restructuring, maybe the authors can come up with something different

but in any case, I strongly believe that the methodology (including a much shortened "theoretical background" section) and the formulation of the steps should be discussed only once. E.g. section 2.2 discusses the controller design theoretically and has a kind of literature review which is then partially repeated in 4.1 where the actual controller re-design is shown. Similarly for the surrogate model sections (2.3 and 4.2) and for the optimization objective functions sections (2.4 and 4.3). I suggest merging these and removing more general discussions for other methods or other possible approaches to sharpen the article and help the reader understand the main contribution. These can be discussed in the discussion section and only once throughout the article.

2. Section 5: I recommend revising this section according to the previous comment. The overlaps and repetitions make it lose its focus and it is difficult to read. E.g.: l 951-964 is the summary of the method and is not needed here, l 997-1000, 1026-1031 have been discussed already earlier in one way or another I think they should only be discussed here. Following the previous comment, I recommend moving in this section all the relevant discussions (uncertainties, possible extensions, alternative methods, limitations), removing summaries and making sure that the points mentioned here have not been discussed earlier.

3. A general comment applicable to most of the sections is that they don't require to have an individual introduction and conclusion. I recommend revising and removing/editing these parts throughout the manuscript to shorten the text and help with the flow. Some examples (not exhaustive): L48-50, 119-127, 243-244, 416-419, 456-459, 613-619, 670-673, 756-760, 849-850, 852-853, etc.

4. There are a lot of sentences throughout the article that state that something will be/was discussed in another section. I recommend removing these as they are too many and make reading difficult. Some examples (not exhaustive): L 116-117, 125, 193-195, 370, 466, 489-491, 516-517, 537, 598, 697-698, 717, 843-845, 852, 868-870, 872-874, etc.

Section 1

5. The first two sub-sections do a good job framing the idea, which is multidisciplinary in nature and novel so it requires context to be explained. My only recommendation would be to discuss also [1] which deals with a similar topic from the perspective of varying prices and fatigue budget management.

6. Section 1.3: I think it is useful but should be considered whether it can be merged with some of the following sections (2 and 4) focusing on the description of the methods. See previous comments.

Section 2: Many of the discussions are too broad and theoretical and in some cases out of scope. At the same time, I found a lot of overlaps with the rest of the manuscript. Many ideas are introduced theoretically and then discussed again in the following sections. As per the previous comments here are some example points to be considered

7. L 248-253: Is this discussion relevant to the topic and the section? Since certification is not touched I suggest removing and discussing it briefly only in the discussion section.

8. L 280-288: Same as before there is no need to explain the IEC standard and refer to the probabilistic approach here

9. Section 2.1.2: Too much detail on the definition of linear damage and DELs and the concept of linear damage, Goodman corrections, etc. L 314-329 could be completely removed and the rest shortened to the derivation of eq. 19 which is relevant for the rest of the work.

10. L 396-401: The different load regions don't need to be discussed here. They are already shown and discussed in the controller design section
11. L420-434: General discussion on surrogate modeling that is out of the scope and overlaps with 4.2. Can be removed or significantly reduced.

For practical reasons, the rest of the comments are focused mostly on the technical side and not on highlighting repetitive parts or discussions that can be removed, which is left to the authors.

Section 3

12. Figure 3: The fonts for the turbine numbering are very small and not readable. I also recommend changing the spacing units to rotor diameters instead of meters.
13. Section 3.1.1: Some more information regarding the simulations is needed. E.g. degrees of freedom used etc. Moreover, some information about the aeroelastic code as the site cited does not include technical details. E.g. Are the aerodynamics calculated with BEM? How is the structural modeling (modal, beam theory)?
14. Section 3.2.1: Show a plot with the wind rose and wind distribution derived from the dataset used. Could be combined with figure 3. This would allow a better understanding of the principal directions and the site-specific wind speed distributions. I think this is important as the whole study depends on the binning of the probabilities but there is no information on how these distributions look anywhere in the article although discussed later (l 823-826). Figure 9 has some of this information but it is much further away and it is too finely binned.
15. L 541-550 too much discussion that is out of the focus of the article, just state the sensors used briefly
16. L578: I am curious about the selection of a constant TI for all ambient wind speeds. According to IEC (conservative) but also from my experience with real data the distribution of TI is correlated to wind speed. Why did you choose not to assume such a correlation? To my understanding, this would not affect the computational time as you already include the TI dimension in the surrogate but would give a more realistic distribution of the mean loads over the wind speed bins.
17. Regarding the conditions considered, I could not find the assumed value of shear for the simulations in the manuscript. Are you using a constant value? Maybe add it to table 2?
18. Section 3.2.2: Is the software also taking into account meandering? How is the superposition of wakes treated? Please clarify briefly.
19. L610: 'This is however neglectably small'. I suggest avoiding such qualitative/subjective statements (eg: clearly, significantly, negligibly, definitely, it is apparent, etc.) and replace with quantitative statements or remove. There are similar expressions throughout the manuscript that should be revised. Other examples: L 799, 801, 805, 831, etc.

Section 4

20. L 686: Using polynomial regression ensures smoothness and continuity by default, which as explained in the previous section is required especially for the gradient-based optimization. The main issue I see with using gradient-based optimizers is the possibility to get stuck on local minima. Did you do something to address this (e.g. varying initial points)?
21. L692: What does "maximum degree of 5" mean in this context? Please clarify

22. L694: If I understand correctly you did 6 simulations of 10 min and used the mean value. Is the assumption here that the 10 and 60-minute load is the same or am I missing something? Can you explain this choice?

23. L 701: "Within this section, two things are presented and discussed." Was it supposed to be a new section here? This sentence seems off (also unnecessary similar to the main comments).

24. L 708: Please explain shortly why you use the training error instead of a test error here. Since the surrogate is going to be probed in any arbitrary value by the optimizer, wouldn't it make sense to show how it performs for inputs outside of the factorial sampling used for training?

25. L 761-764: I understand that the focus of the work is not on the optimization algorithms, but some more information on the application would be useful for the general understanding. What starting values were used? How many iterations did it require to converge? How fast was it?

26. Figure 6,7: Add in the caption and maybe in the plot the units for the y axes. Meaning, explain it is normalized and state the values used for normalizing. Additionally, it seems the factors for normalizing are randomly chosen. E.g. why not make the power ND with the nominal value (100%) so that the levels of down-regulation are directly seen in the plot? Similarly for the loads.

27. L 773-776:  IEC class IA is the highest turbulence class. Is this a typo? In general the sentences "Due…Sect.2"  seem to have some text missing or be misplaced. Please revise for clarity

28. Figure 8: This plot is difficult to read due to small fonts and graphic size. I suggest making the turbine markers bigger so that the colour differences are distinguishable. Also, the fonts of the number next to the turbine can be bigger and outside of the turbine circles.

29. Table 3: what does "with reference from turbine 4" mean? Are these relative damage values? If so the energy has to be also relative. Please clarify in the caption and adjust the table accordingly

30. Table 4: what does "relative damage and energy production compared to operation without derating" mean? Are these values for each turbine relative to its own baseline or to some other turbine? If so, how come the damage values of turbine 4 are the same as table 3? Please clarify in the caption and adjust the table accordingly

31. The previous comments for the relative values are valid in general for the text in section 4. When interpreting the results (e.g. l 782, 783, 789, 790, 803) the notation of values with decimals (e.g. 0.7) are used interchangeably for relative differences and for absolute damage values (which also scale to 1). This can be confusing. My suggestion is to change the notation throughout the manuscript and state percentages when talking about relative values and absolute numbers for the damage.

32. L 822-826: In figures 10b,c and 11b,c it seems that the highest damage/frequency for turbine 4 comes from the wind direction 180. Looking at the previous plots this sector seems to have a low probability of occurrence and low wind speed magnitudes in general. I would expect the sector 200-315 deg to be the most influential. Please comment on this.

33. L 839-845, 891-895: These are not part of the discussion of the results, are overlapping with other sections and can be removed.

34. Figures 12 and 13: As per previous comment I suggest using percentages for relative values to distinguish with the absolute values mentioned in L.860-863, 884, etc.

35. L 913: 'The numbers are actually valid for a full wind farm' What does this mean? Including also other farm-related costs? Please clarify

36. Energy and financial benefits of lifetime extension l805, 873-877, sections 4.4 :
I think this calculation of extra energy production through lifetime extension (and subsequently revenue) has a lot of underlying assumptions. Subsidies are over after the nominal life (usually even earlier) and the selling prices would be reduced or be subjected to the volatility of the market. There are many other factors to lifetime extension like permitting, land costs, inspections/certification and wear and tear of components besides the main ones (bearings,

actuators, gearbox, etc.). Other failure modes, like leading edge erosion, could also lead to either reduced power production, extended downtime, or even make the lifetime extension financially infeasible in total. Additionally, the assumption that all blades would be replaced for all turbines in a farm to operate for a few years more is not realistic. I understand that this is a first approach and a proof of concept but I think these should be clearly stated and discussed more (briefly mentioned in l946-947) as the results can be misleading for actual decision-making.

Section 5

37. I think the assumptions of constant price and costs for the whole operational time including LTE has to be discussed here. As mentioned in the previous comment I think these are the highest contributors to uncertainty and should be emphasized to give the correct perspective for the monetary results presented here.
38. Similarly, inter-annual variability is an important factor of uncertainty based on the assumptions of the study and has to be discussed. Due to the non-linear relationship between loads and conditions, it is hard to predict what the effect would be. A good starting point for the topic could be [2].
39. L 1076-1077: This is not clear to me. Are you referring to the mismatch of actual conditions to the mean wind distribution? Since the wind distributions are pre-defined in the optimization logic how would the actual conditions change the result with the current approach? Please clarify

**Minor corrections**

- L259: lateron
- L603: bins.D?
- L 775: however?
- L 776: introducing?
- L 785: for instance?
- Tables 3 and 4: Adjust all the values in the table to the same decimal. I.e. 0.7-->0.700 etc
- L1050 many-objective→multi-objective?

**References**

[1] Kölle, K., Göçmen, T., Eguinoa, I., Alcayaga Román, L. A., Aparicio-Sanchez, M., Feng, J., Meyers, J., Pettas, V., and Sood, I.: FarmConners market showcase results: wind farm flow control considering electricity prices, Wind Energ. Sci., 7, 2181–2200, https://doi.org/10.5194/wes-7-2181-2022, 2022.

[2] Pryor, S. C., Shepherd, T. J., and Barthelmie, R. J.: Interannual variability of wind climates and wind turbine annual energy production, Wind Energ. Sci., 3, 651–665, https://doi.org/10.5194/wes-3-651-2018, 2018.

---

## Author Comment (AC1)

**Final author comments as response to reviewers on the original manuscipt**

**From wind conditions to operational strategy: Optimal planning of wind turbine damage progression over its lifetime**

Niklas Requate, Tobias Meyer, Rene Hoffmann

April 2023

We kindly thank the reviewers for their detailed and valuable feedback. Both reviewers give positive feedback on the relevance of the results, but have concerns about the readability due to length and structure. They suggest reworking the structure for a better readability and to shorten the amount of text.

We are working on a revised manuscript where we have already implemented their suggestions almost completely and which we will upload after the discussion has finished. Below, you find our detailed reply to each comment. Since both reviewers gave similar suggestions for restructuring the sections, we respond to both together. Afterwards, we reply to specific comments of both reviewers. Our replies are written in red below each reviewer comment. In our "Response to the structure and the readability", we describe how the revised manuscript is structured and what major changes were done to improve the clarity of our work.

**Comments on the structure**

**Reviewer 1**

**General Comments**

This is an interesting article with a novel approach to wind turbine operation showing an alternative to the single-dimensional current operational paradigm. The authors have put a lot of thought and effort into the manuscript addressing a multidisciplinary topic. It has a thorough literature review, the research question is framed sufficiently, the methods used are explained adequately, the language used in terms of syntax and typos is in general good and the topic is relevant to the community and the scope of the journal. I have concerns about some of the assumptions used, especially for the economical evaluation of lifetime extension, but I reckon that this is a proof of concept and I suggest that these are discussed rather than directly addressed. My major concern

is about the readability and clarity of the manuscript. The article is too long with a lot of repetitions and overlaps between the sections making it cumbersome to understand and difficult to navigate through. Moreover, the authors explain extensively methods that are not in the scope of the article or give much more detail than required in places. I think the article has to be restructured, sharpened and significantly decreased in length to be suitable for publication as a journal article. I believe the manuscript is relevant and should be published after addressing the issues mentioned here through a major revision. Find more specific comments below.

**Specific comments (Part 1)**

1. Sections 1.3, 2 and 4 have a lot of overlaps, repetitions and some general discussions being more than 26 pages combined. I suggest merging them and discussing each topic only once. E.g.: Show the general method briefly (similar to section 1.3) as an introduction and merge there sections 2, 4.1 and 4.2. Then make a new section only with the results of the use cases (sections 4.3, 4.4). This is an example of possible restructuring, maybe the authors can come up with something different but in any case, I strongly believe that the methodology (including a much shortened "theoretical background" section) and the formulation of the steps should be discussed only once. E.g. section 2.2 discusses the controller design theoretically and has a kind of literature review which is then partially repeated in 4.1 where the actual controller re-design is shown. Similarly for the surrogate model sections (2.3 and 4.2) and for the optimization objective functions sections (2.4 and 4.3). I suggest merging these and removing more general discussions for other methods or other possible approaches to sharpen the article and help the reader understand the main contribution. These can be discussed in the discussion section and only once throughout the article.

**Reviewer 2**

My major comment is about the clarity of the paper. First of all, it is cumbersome to understand. Secondly, It is too long with a lot of overlapping descritipions which makes it difficult to read. It is a journal paper not a thesis! I suggest that the article needs to be re-organized and decreased the length. Additionally, there are lot of sentences in this paper which state "based on assumption", "we assume". It can be subjective or over simplify the problem, which make your conclusions less concrete and solid. I believe the manuscript should be published only after a major revision by addressing the issues mentioned in this review.

Section 2.2 to 2.4 are similar to section 1.3. The difference is it includes more detailed theoretical description and literature study. So maybe you could merge them and shorten the content?

**Response to the structure and the readability**

We appreciate the concrete proposals for restructuring and have implemented them. The previous division of sections 1.3, 2 and 4 has been dissolved. The new structure is as follows:

1. Introduction
    1.1. State of the art
    1.2. Objectives
    1.3. Methodology
    1.4. Outline of the remaining paper
2. Theoretical background
    2.1. Long-term fatigue damage progression and energy production depending on external conditions and operational planning
    2.2. Relationship between fatigue damage and damage equivalent laod (DEL)
3. Definition of example system and prerequesites
    3.1. System boundaries for application example
    3.2. Adaptable real-time controller of the wind turbine
    3.3. Surrogate models for damage progression and energy production
4. Method for optimal long-term planning: VIOLA
    4.1. Optimization process for operational strategies
    4.2. Select economically best operational lifetime-planning strategy
5. Discussion
6. Conclusion and Outlook

In section 1.3, the 4 steps are now only very briefly summarized. They have also been split into prerequisites for the new method (steps 1 and 2) and the actual method (steps 3 and 4). The rest of the paper structure is also structured based on this division. This has already removed many of the unnecessary and sometimes redundant transitions.

In order to improve readability and clarity, we improved focus on the method itself. To this end, we removed one use case entirely (Section 4.3.1 "Levelled farm damage" in the old manuscript). Within this work, we now restrict ourselves to one turbine within a farm as application example. This omitted use case is now mentioned in the Outlook. We still consider addressing a full wind farm as highly relevant for the future.

We also removed a lot of discussions in the subsections and added the most relevant parts into the "Discussion"-chapter. In total, the revised manuscript will be reduced by

more than 30% , bringing it from 44 pages to approximately 34 pages in the main part (excluding references, list of symbols and author statements).

**Reviewer 1**

**Summary**

The article suggests and evaluates a novel operational method for wind turbines within a wind farm. The long-term fatigue consumption and energy production of each turbine is managed individually by adapting the power output level according to wind conditions to reach lifetime objectives. The motivation for such an adaptation is explained along with all the steps of the process including the controller re-design, the surrogate modeling and the relevant optimization techniques. The results are demonstrated with example use cases, showing the potential applications for equalizing the fatigue consumption among the turbines and a Pareto analysis on fatigue consumption reductions which are then converted to lifetime extension potential and eventually translated to potential economic benefits.

**Specific comments (part 2)**

2. Section 5: I recommend revising this section according to the previous comment. The overlaps and repetitions make it lose its focus and it is difficult to read. E.g.: l 951-964 is the summary of the method and is not needed here, l 997-1000, 1026-1031 have been discussed already earlier in one way or another I think they should only be discussed here. Following the previous comment, I recommend moving in this section all the relevant discussions (uncertainties, possible extensions, alternative methods, limitations), removing summaries and making sure that the points mentioned here have not been discussed earlier.

We revised section 5 according to your suggestion. The summary is removed. All the relevant discussion from previous chapters are moved to this part. You find some additional reply on section 5 later at comments 36. and 37.

3. A general comment applicable to most of the sections is that they don't require to have an individual introduction and conclusion. I recommend revising and removing/editing these parts throughout the manuscript to shorten the text and help with the flow. Some examples (not exhaustive): L48-50, 119-127, 243-244, 416-419, 456-459, 613-619, 670-673, 756-760, 849-850, 852-853, etc.

We removed most of these individual introductions and conclusions. Some of these have also been eliminated due to restructuring. Some are rewritten into a transition to the next step of the process.

4. There are a lot of sentences throughout the article that state that something will be/was discussed in another section. I recommend removing these as they are too many

and make reading difficult. Some examples (not exhaustive): L 116-117, 125, 193-195, 370, 466, 489-491, 516-517, 537, 598, 697-698, 717, 843-845, 852, 868-870, 872-874, etc.

We removed all of the mentioned references to other sections. Some of these were also resolved by the restructuring. We also looked out for similar text parts and removed them if not absolutely required for understanding.

**Section 1**

5. The first two sub-sections do a good job framing the idea, which is multidisciplinary in nature and novel so it requires context to be explained. My only recommendation would be to discuss also [1] which deals with a similar topic from the perspective of varying prices and fatigue budget management.

Thank you for pointing out the additional source which is certainly relevant for this context. We now discuss it as part of the "State-of-the-art" and also mention it in the outlook.

6. Section 1.3: I think it is useful but should be considered whether it can be merged with some of the following sections (2 and 4) focusing on the description of the methods. See previous comments.

This was a major part of the restructuring. Section 1.3 now only briefly introduces the four steps without many details (see comment on restructuring).

**Section 2**

Many of the discussions are too broad and theoretical and in some cases out of scope. At the same time, I found a lot of overlaps with the rest of the manuscript. Many ideas are introduced theoretically and then discussed again in the following sections. As per the previous comments here are some example points to be considered

We agree that a lot of ideas were discussed in too much detail. Thus, the former section 2 does not exist anymore and has been subdivided into the new section 2 for the theory of fatigue damage, section 3 system boundaries and the implementation of prerequisites and section 4 for the optimization process and economic evaluation. We see section 2 as a required theoretical introduction to the long-term fatigue progression. Within the other parts, we shortened the amount of theoretical discussions significantly.

7. L 248-253: Is this discussion relevant to the topic and the section? Since certification is not touched I suggest removing and discussing it briefly only in the discussion section.

We do as you recommend and only briefly point out the major difference to the certification in the theory-section 2.

8. L 280-288: Same as before there is no need to explain the IEC standard and refer to the probabilistic approach here

We removed this discussion to increase clarity and readability. We mention the probabilistic approach in section 5.

9. Section 2.1.2: Too much detail on the definition of linear damage and DELs and the concept of linear damage, Goodman corrections, etc. L 314-329 could be completely removed and the rest shortened to the derivation of eq. 19 which is relevant for the rest of the work.

The mentioned lines are completely removed. The other parts built on one another for the derivation of Eq. 19. Therefore, we decided not to shorten this part further.

10. L 396-401: The different load regions don't need to be discussed here. They are already shown and discussed in the controller design section.

Sections 2.2 and 4.1 are merged as suggested. Therefore, the discussion of load regions is only in one place, but shortened giving less details. Because of the differing influence on loads in both regions and their relevance for optimization results, some discussion is required in our opinion.

11. L420-434: General discussion on surrogate modeling that is out of the scope and overlaps with 4.2. Can be removed or significantly reduced.

The discussion is significantly reduced. Surrogate models are now considered as required prerequisite for the optimization process. For this reason, most details on our implementation are removed. Sections 2.3 and 4.2 are also merged as suggested (see "Response to the structure and the readability" above). Therefore, the overlap is removed.

For practical reasons, the rest of the comments are focused mostly on the technical side and not on highlighting repetitive parts or discussions that can be removed, which is left to the authors.

The restructuring removed overlapping sections mostly. In addition, we went over the paper for another critical shortening. During this, many additional sections were removed or merged.

**Section 3**

12. Figure 3: The fonts for the turbine numbering are very small and not readable. I also recommend changing the spacing units to rotor diameters instead of meters.

We did these changes as suggested.

13. Section 3.1.1: Some more information regarding the simulations is needed. E.g. degrees of freedom used etc. Moreover, some information about the aeroelastic code as the site cited does not include technical details. E.g. Are the aerodynamics calculated with BEM? How is the structural modeling (modal, beam theory)?

More information and two more sources are added in order to clarify that MoWiT indeed does compute aerodynamics with BEM and is a near-equivalent to Bladed.

14. Section 3.2.1: Show a plot with the wind rose and wind distribution derived from the dataset used. Could be combined with figure 3. This would allow a better understanding of the principal directions and the site-specific wind speed distributions. I think this is important as the whole study depends on the binning of the probabilities but there is no information on how these distributions look anywhere in the article although discussed later (l 823-826). Figure 9 has some of this information but it is much further away and it is too finely binned.

The figure of the wind rose has been moved to the section were site-specific wind conditions are introduced (new section 3.1). The fine binning is kept, because it is also used to compute the total damage and energy production. To clarify the binning process for optimization, a figure on the additional binning into wind and TI bins is added.

15. L 541-550 too much discussion that is out of the focus of the article, just state the sensors used briefly

The discussion part was removed from this section and is covered in the "Discussion" section 5.

16. L578: I am curious about the selection of a constant TI for all ambient wind speeds. According to IEC (conservative) but also from my experience with real data the distribution of TI is correlated to wind speed. Why did you choose not to assume such a correlation? To my understanding, this would not affect the computational time as you already include the TI dimension in the surrogate but would give a more realistic distribution of the mean loads over the wind speed bins.

You are right, the assumption of a low constant ambient TI is not realistic. Since the main focus was on the development of the optimization method, we did not place such a high value on the selection of the specific input conditions. We addressed this issue now by creating new results using TI from the IEC class B Weibull distribution, depending on wind speed. Here, we use the 50% quantile. Such TI values are usually to high for offshore wind farms but still realistic onshore. These new values and derived values were used for all subsequent sections as well. In effect, the entire application example was re-computed. New plots use a different color map due to changed corporate identify colors. These new colors are optimized for gradually increasing luminosity values, which also makes it easier to distinguish individual plots when printed in black and white.

17. Regarding the conditions considered, I could not find the assumed value of shear for the simulations in the manuscript. Are you using a constant value? Maybe add it to table 2?

A constant value for all other parameters except wind speed and TI was used (The power law shear exponent is 0.2). We added a statement on that in Section 3.1.1.

18. Section 3.2.2: Is the software also taking into account meandering? How is the

superposition of wakes treated? Please clarify briefly.

We added a clear statement that wake meandering is not taken into account. The resulting model uncertainty is discussed in a later section. Model uncertainties can be removed by considering more influences on the loads and improved models. For this paper, the optimization method and the general process was the focus.

19. L610: 'This is however neglectably small'. I suggest avoiding such qualitative/subjective statements (eg: clearly, significantly, negligibly, definitely, it is apparent, etc.) and replace with quantitative statements or remove. There are similar expressions throughout the manuscript that should be revised. Other examples: L 799, 801, 805, 831, etc.

We agree to this comment and removed such wording in the mentioned examples and throughout the rest of the text.

**Section 4**

20. L 686: Using polynomial regression ensures smoothness and continuity by default, which as explained in the previous section is required especially for the gradient-based optimization. The main issue I see with using gradient-based optimizers is the possibility to get stuck on local minima. Did you do something to address this (e.g. varying initial points)?

We add a statement regarding local minima in Section 4: "As starting values, the reference strategy with 100% power production at each turbine was always used, which is always a non-optimal but feasible solution. All optimization runs show plausible results in terms of an improved relationship between energy increment and damage increment. For this reason, no explicit variations of the starting values were required to check for convergence to local minimal.".

21. L692: What does "maximum degree of 5" mean in this context? Please clarify

In the text, we clarified that this refers to the maximum order of the polynomial.

22. L694: If I understand correctly you did 6 simulations of 10 min and used the mean value. Is the assumption here that the 10 and 60-minute load is the same or am I missing something? Can you explain this choice?

Yes, we hereby refer to the standard minimum number of 6 seeds per wind condition and compute the short term DEL from that. The uncertainties through the number of seeds is definitely discussable. We clarified our approach and also added a (newly published) source on the required number of seeds for fatigue calculations.

23. L 701: "Within this section, two things are presented and discussed." Was it supposed to be a new section here? This sentence seems off (also unnecessary similar to the main comments).

The sentence is removed as part of the revision of the entire section

24. L 708: Please explain shortly why you use the training error instead of a test error here. Since the surrogate is going to be probed in any arbitrary value by the optimizer, wouldn't it make sense to show how it performs for inputs outside of the factorial sampling used for training?

We now assume the surrogate model to be a prerequisite and thus omit most details about our implementation. Due to this, the discussion on the accuracy of the fits was significantly shortened. Our main point, which we clarified, is that these models are sufficient to be used by the optimization method.

25. L 761-764: I understand that the focus of the work is not on the optimization algorithms, but some more information on the application would be useful for the general understanding. What starting values were used? How many iterations did it require to converge? How fast was it?

We added a text passage regarding these questions.

26. Figure 6,7: Add in the caption and maybe in the plot the units for the y axes. Meaning, explain it is normalized and state the values used for normalizing. Additionally, it seems the factors for normalizing are randomly chosen. E.g. why not make the power ND with the nominal value (100%) so that the levels of down-regulation are directly seen in the plot? Similarly for the loads.

We clarify in the caption and in the units how the values are normalized. Now, it is normalized to nominal power.

27. L 773-776: IEC class IA is the highest turbulence class. Is this a typo? In general the sentences "Due...Sect.2" seem to have some text missing or be misplaced. Please revise for clarity

This sentence was removed because its meaning was unclear and not really relevant at that point.

28. Figure 8: This plot is difficult to read due to small fonts and graphic size. I suggest making the turbine markers bigger so that the colour differences are distinguishable. Also, the fonts of the number next to the turbine can be bigger and outside of the turbine circles.

Figure 8 is completely removed from this paper as it was part of the use case on "levelled farm damage" which was removed (See "Response to the structure and the readability").

29. Table 3: what does "with reference from turbine 4" mean? Are these relative damage values? If so the energy has to be also relative. Please clarify in the caption and adjust the table accordingly 30. Table 4: what does "relative damage and energy production compared to operation without derating" mean? Are these values for each turbine relative to its own baseline or to some other turbine? If so, how come the damage values of turbine 4 are the same as table 3? Please clarify in the caption and adjust the table accordingly

31. The previous comments for the relative values are valid in general for the text in section 4. When interpreting the results (e.g. l 782, 783, 789, 790, 803) the notation of values with decimals (e.g. 0.7) are used interchangeably for relative differences and for absolute damage values (which also scale to 1). This can be confusing. My suggestion is to change the notation throughout the manuscript and state percentages when talking about relative values and absolute numbers for the damage.

This part is also completely removed was part of the use case on "levelled farm damage".

32. L 822-826: In figures 10b,c and 11b,c it seems that the highest damage/frequency for turbine 4 comes from the wind direction 180. Looking at the previous plots this sector seems to have a low probability of occurrence and low wind speed magnitudes in general. I would expect the sector 200-315 deg to be the most influential. Please comment on this.

With the change of the ambient TI value, the damage distributions also have changed slightly. In general, high damage is not only induced in the main wind direction but also from the high wake-induced turbulence. We hope to have clarified this in the explanation of the text.

33. L 839-845, 891-895: These are not part of the discussion of the results, are overlapping with other sections and can be removed.

Both paragraphs are removed as suggested.

34. Figures 12 and 13: As per previous comment I suggest using percentages for relative values to distinguish with the absolute values mentioned in L.860-863, 884, etc.

We also changed the relative values to percentage values for clarity.

35. L 913: 'The numbers are actually valid for a full wind farm' What does this mean? Including also other farm-related costs? Please clarify

The formulation was misleading. We changed it to "The financial estimations refer to an entire wind farm".

36. Energy and financial benefits of lifetime extension l805, 873-877, sections 4.4 : I think this calculation of extra energy production through lifetime extension (and subsequently revenue) has a lot of underlying assumptions. Subsidies are over after the nominal life (usually even earlier) and the selling prices would be reduced or be subjected to the volatility of the market. There are many other factors to lifetime extension like permitting, land costs, inspections/certification and wear and tear of components besides the main ones (bearings, actuators, gearbox, etc.). Other failure modes, like leading edge erosion, could also lead to either reduced power production, extended downtime, or even make the lifetime extension financially infeasible in total. Additionally, the assumption

that all blades would be replaced for all turbines in a farm to operate for a few years more is not realistic. I understand that this is a first approach and a proof of concept but I think these should be clearly stated and discussed more (briefly mentioned in l946-947) as the results can be misleading for actual decision-making.

The aim of our method is to exploit the full load-bearing capacity of one component. In Sections 4.1 and 4.2, we now discuss the effect of optimization wrt one failure mode on all other failure modes in more detail. We will address the combination of multiple failure modes into one operating strategy in a future paper. On a technical note, our method would also be capable using a turbine power boost to increase energy yield, revenue, but also accelerate ageing of components. Also, the approach is capable of optimizing the strategy with respect to different price ranges which could be coupled to the wind frequency. Combined with possibilities of a power boost, this would be a suitable way to address limited land lease periods or other boundary conditions that inhibit lifetime extension. However, our implementation of an adaptable turbine controller, which is a prerequisite but not a focus in this paper, only allows for derating but not for such power boost. overall, we agree that there are many other factors influencing the circumstances under which it is worthwhile to adjust the operational strategy. We will give greater consideration to this aspect in the discussion.

**Section 5**

37. I think the assumptions of constant price and costs for the whole operational time including LTE has to be discussed here. As mentioned in the previous comment I think these are the highest contributors to uncertainty and should be emphasized to give the correct perspective for the monetary results presented here.

We will make sure, that we emphasize these assumptions more clearly. We will start the introduction with this statement to put the results into a general perspective. Within this work, the focus lies on the technical optimization approach. In future work, we aim at addressing the varying prices in more detail.

*"With the application example, we aimed at showing that all four identified steps build on one another and how the process can be used for an effective distribution of damage over the entire lifetime of a turbine. This way the interaction of the inputs, such as control setpoints, environmental conditions, damage progression, energy production and also economic value becomes clear. The considered application example mainly illustrates that the mathematical optimization method is applicable for creating operational strategies and how the method can be applied to exploit the full load-bearing capacity of one component and to increase the value of the considered system. The mathematical optimization is built on the assumptions of the used models and their input data. It finds the best operational strategy under these assumptions in a deterministic way. Thus, the solution also includes the uncertainties resulting either from model inaccuracies or from uncertain assumptions in the input data. This inherent limitation must always be taken into account when evaluating and discussing the results. For this reason, we have taken*

*great care to describe the required prerequisites in detail. In addition, we have limited ourselves to the technical level first when performing the optimization (step 3). The consideration of economic factors is subject to a high degree of uncertainty due to the uncertainty of future electricity prices and other influences which are not considered. The selection of an operational strategy must always be made for a specific application, taking into account the inherent uncertainties and risk. The selection process applied to the considered system in this work mainly aims at showing how the economic aspect can be taken into account and that an intelligent operational strategy can lead to economic profit."*

Also, the specific assumptions and limitations of the economic evaluation will be discussed in more detail.

38. Similarly, inter-annual variability is an important factor of uncertainty based on the assumptions of the study and has to be discussed. Due to the non-linear relationship between loads and conditions, it is hard to predict what the effect would be. A good starting point for the topic could be [2].

We agree that the inter-variability is one factor of uncertainty. We will state this clearly in the discussion. We will also address inter-annual variability of the wind a future work about the combination of failure modes.

39. L 1076-1077: This is not clear to me. Are you referring to the mismatch of actual conditions to the mean wind distribution? Since the wind distributions are pre-defined in the optimization logic how would the actual conditions change the result with the current approach? Please clarify

We will clarify this. With this paragraph, we aimed to relate back to the approach of reliability-(adaptive) control. Even with a perfect planning, there will be deviation from it due to the individual progress of each turbine component and the forecasting uncertainties. Therefore, continuous adaptions of the operation depending on the current status of the turbine and advanced planning would result in a closed-loop control of the reliability (The operating stage in Figure 1 which is not part of this work)

**Minor corrections**

• L259: lateron • L603: bins.D? • L 775: however? • L 776: introducing? • L 785: for instance? • Tables 3 and 4: Adjust all the values in the table to the same decimal. I.e. 0.7→0.700 etc • L1050 many-objective→multi-objective?

All the minor corrections are incorporated or the part of text is removed entirely.

**Reviewer 2**

The authors have done a good job for presenting a novel method for operating wind farm, in which

the long-term fatigue consumption and energy production of each turbine are managed individually to reach the objectives. The authors addressed that relevant deterministic external conditions and real-time controller setpoints influence the damage progression with equal importance. The authors use case studies/examples to illustrate the conclusions made in the paper, which was somehow effective, as it gave examples of the benefits of the potential applications for equalizing the fatigue consumption among the turbines and the impact can have on potential economic benefits

**General Comments**

This is an interesting article with a novel approach to operate wind turbines. It has a complete literature studies, the research question is sufficient, the methods are explained by using good terminology. the topic is relevant to the community.

**Specific comments**

Line 167 , ... for setpoints for the real-time ...→ ... for setpoints of the real-time ...

Line 183: means -> ways

Both previous comments have been resolved by revising section 1.3.

Line 208: Even with the explanations in the footnote (2), It is still unclear to me which one is variable and which one is parameter. Because the author only metioned they are separated by semicolon without mentioning which one is in front.

We address this comment by clarifying the explanation in the footnote and giving an example: "Note that in our notation we distinguish between inputs and parameters of the defined function. Parameters are assumed to be fixed for a specific use case. They are separated by a semicolon, where the function inputs are in front of the semicolon. If additional parameters exist but are not important for a certain passage, we omit them to improve readability and replace them with a central dot ($\cdot$). So for $D_{fm}(\tau^{ref}; \bar{u}^{ref}, \cdot)$, $\tau^{ref}$ is an input, $\bar{u}^{ref}$ is a set of parameters and $\cdot$ denotes that additional parameters are omitted."

Figure 2. It is not clear to me what do the dash lines represent when I read the figure without trying to look for the explanation in the contexts.

We address this comment by adding the explanation to the legend. We refrain from mentioning it in the legend in order not to overload the figure.

L242: sound?

The "sound basis" was meant in the sense of "solid foundation" or "correct". However, this sentence is removed in the revised paper.

L248 to L254: I don't think this discussion is relevant to the paper.

We do as you recommend and only briefly point out the major difference to the certification in the theory-section 2.

(same answer as to comment 7. of reviewer 1)

L259 - L260: ... The sets X and U will lateron be clearly defined based on the system boundaries ... It is better to describe directly here in order to improve the readability. Beside this, "lateron" should be "later on".

We address this comment by defining both of them at their first occurence.

Trying to unify the usage of the mathmatical symbol. One example, In equation (1): time increament is represented by $\Delta\tau$, in L260, you use $\Delta t$ to denote time increment.

This distinction is intentional in order to distinguish different time scopes. In order to clarify these differences, a graphic has been introduced that explicitly explains the different time scopes, their transition, and usage at different stages.

Section 3.1: this section title has no relation to the sub-sections. Please consider to modify it.

Due to the restructuring, many section titles were also adjusted. The new section 3.1 now describes the system boundaries of the entire application example. The new section 3.1.1 corresponds to the previous section 3.1. We have slightly adjusted the title but feel that it is generally appropriate. These are the new titles for the definition of the example:

*3. System boundaries for application example*

*3.1. Modelling of single turbine and its system boundaries*

*3.2. Wind farm setup: From surrounding system to considered wind turbine*

section 3.11: More information regarding the simulations is needed. The cited reference does not include any technical details. which controller is used?

More information and two more sources are added in order to clarify that MoWiT indeed does compute aerodynamics with BEM and is a near-equivalent to Bladed.

(same answer as to comment 13. of reviewer 1)

The controller is described in new section 3.2 in more detail. This is now explicitly clarified.

L534: The authors mentioed that the wake effects are covered only through an increase of turbulence intensity. Actually, the wake meandering has even more affects on the fatigue loads on the downstream turbines. Without included the wake mandering make the conclusion less solid.

We added a clear statement that wake meandering is not taken into account. The resulting model uncertainty is discussed in a later section. Model uncertainties can be

removed by considering more influences on the loads and improved models. For this paper, the optimization method and the general process was the focus.

(same answer as to comment 18. of reviewer 1)

L544: "bm", what is this? bending moment? so far until L544, this abbreviation is not defined. later the abbreviation appears in table 1.

It indeed stands for bending moment. We address your comment by adding a footnote at its first occurrence. The abbreviation is required because the term is extensively used throughout the text, in figure labels and in formula symbol subscripts. Due to this, we feel it is justified to introduce an abbreviation.

L544-L545: The correct statement should be the variation of the edgewise bending moment is driven by gravity loads. The fluctuation of flapwise bending moment is strongly influenced by the turbulence. But the controller setpoints plays a major role. But you did not mention this.

You are right with your statement. This is what we wanted to express. We clarified this at the first occurence (now part of section 3.1.1) and later in the text. Our revised passage reads as follows:

*"All these loads can be considered as representatives for the fatigue accumulation of different components that can be influenced by the wind turbine controller and the environmental conditions in different ways. While the tower and the flapwise bending moment are more strongly influenced by turbulence, the variations in the edgewise bm are driven by gravity loads dependent on the rotor speed, i.e. the controller and the wind speed."*

L578: the TI is set to 5%? Is this realistic? According to my knowledge and based on IEC standard, this value should be correlated to the mean wind speed. Why do the authors use a constant value?

You are right, the assumption of a low constant ambient TI is not realistic. Since the main focus was on the development of the optimization method, we did not place such a high value on the selection of the specific input conditions. We addressed this issue now by creating new results using TI from the IEC class B Weibull distribution, depending on wind speed. Here, we use the 50% quantile. Such TI values are usually to high for offshore wind farms but still realistic onshore. These new values and derived values were used for all subsequent sections as well. In effect, the entire application example was recomputed. New plots use a different color map due to changed corporate identify colors. These new colors are optimized for gradually increasing luminosity values, which also makes it easier to distinguish individual plots when printed in black and white.

(same answer as to comment 16. of reviewer 1)

L582: You need the reference for FOXES, both the footnotes (3) and the cited paper (Schmidt et al., 2021) that you used do not contain information about this code. Please cite the correct paper.

The software FOXES has been published open source in the meantime. We cite the repository now and refer to the documentation. There is no paper or document which mainly focuses on introducing the software.

L583: ... In this case, only a small part of the software is used ... This sentence is not professional and needs to be revised.

This sentence is removed.

L610: 'This is however neglectably small'. I suggest avoiding such qualitative statements and replace with quantitative statements or just don't use. There are similar expressions throughout the paper.

We agree to this comment and remove such wording in the mentioned case and throughout the rest of the text.

(similar answer as to comment 18. of reviewer 1)

section 4.1: The title of section 4.1 is exactly the same as section 2.2, this time with a bit more information compared to section 2.2, Because only now, it is really linked to a concrete controller design and description. So my suggestion is to re-organize the structure, remove the overlapping and make the paper more concise and readable. For now, it is really difficult to read through. The reader can easily lost their focus. (The same for section 4.2, 4.3 and 4.4, please consider to revise)

This comment is addressed by the full restructuring of the paper (See "Response to the structure and the readability"). The mentioned sections are merged as you and reviewer 1 suggested.

Section 4.1: For the derating strategy used in the test wind farm, have you considered the minimum thrust coefficient derating strategy as describe in this reference (Meng et al., 2020: The effect of minimum thrust coefficient control strategy on power output and loads of a wind farm) and what is the results compared to your 3 derating methods? Please clarify this.

We clarified this. Currently, we only use the constant-lambda strategy. The major focus was to create a single setpoint for the derating of power which can later be used for optimization. This is hopefully made more clear by a full revision of the section about the adaptable controllers (now section 3.2).

Some general comment on the formulas: Try to avoid use word abbreviation in the formulas, except for superscript or lowerscript, because it reduces the readability.

We reduced the total number of word abbreviations but in many cases they are necessary to clarify the exact scope of a symbol, e.g., total damage value D and reference value $D^{ref}$. We also cleaned up typesetting so that abbreviations can now more clearly be distinguished from multiple individual letters. To some extent, it is our personal style to use some of these abbreviation because it is more understandable than using a number or just one letter.

L682-685: Diffcult to understand. Please consider to revise the sentences.

We revised the entire section to improve clarity.

L689: is $\rightarrow$ are,

Is changed.

L692: "... degree of 5 ..." is this the order of the polynomial? Please clarify this.

It is the maximum order of the polynomial, yes. This was also clarified in the text.

(same answer as to comment 21. of reviewer 1)

Table 2: please clarify the notation that you used in the column of turbulence intensity.

We added an explanation: *"While wind speed and power are sampled equidistantly, the sampling of the TI values is selected so that the distance between the samples increases exponentially, as indicated by the formula in table 2".*

L727-728: "...The load reduction of the edgewise bm directly corresponds to the reduction in rotor speed..." what do you mean? Load reduction (fatigue) of the edgewise can relate to the less fluctuation in rotor speed. But it can not reduce the rotor speed.

We rewrote the sentece to clarify this:

*"The reduction in $DEL_{edge}^{st}$ depending on $\delta_P = 100\%$ directly relates to the lower rotational speed through the control setpoints at each wind speed. Thus, it has a stronger effect at 90 % and 80 % when the rotor speed is lowered by a higher amount than the generator torque to achieve the power setpoint."*

L732: In the simulation setup, L578, you have mentioned that the TI is set to 5% for all wind speed, but here it is 11.3% for all wind speed. Probably here, the 11.3% TI is the wake induced turbulent Intensity? Please clarify this.

For the input conditions to the aero-elastic simulations on the seconds scope, we do not distinguish between wakw-induced TI and standard TI. In the current study, the wake just increases the TI input into the surrogate model. Therefore, TI values are relevant in a range depending on wind speed and the selection of fix TI in this section is just one example. Nevertheless, we changed the fix value to 16% because we also increase the ambient TI for the results.

Figure 6, the normalization approach is not clear to me. for example, the power is not normalized by 100% derating case. The normalized loads are also strange. Please check and modified the normalization procedure.

We clarify this by mentioning it in the caption and changing the label.

L773-776: " ... Due to the low .... explanation in Sect. 2. ..." This sentence is not understandable. please revise it.

The whole section on the example of "Levelled Farm damage" was removed (See main description of restructuring). Also, the mentioned sentence is deleted.

Figure 8: please consider to improve the readability, for example, increase the size of the circle and change the location of the number in the plot.

Figure 8 is completely removed from this paper as it was part of the use case on "levelled farm damage" which was removed (See "Response to the structure and the readability").

(same answer as to comment 21. of reviewer 1)

Table 3: You mentioned " total damage and energy production with reference from turbine 4". Based on my understanding, the values for turbine No.4 should be 1, right? And why the energy production is still with units?

Should not it also be a normalized value? Please clarify this.

Table 4: This table is also not clear to me. very confusing. So this need to be clarified and revised accordingly.

The whole section on the example of "Levelled Farm damage" was removed, and thus also the tables and the results (See "Response to the structure and the readability"). The comments will be kept in mind for future work.

figure 9: Please consider to use "probability" to replace "frequency". Because you are discussing the probability distribution of wind condition. Please also check the whole manuscript and change them accordingly if it applys.

Quite frankly, we're not happy with the term *frequency* here as well, since it can create confusion with the exact same term used for vibration signals.

However, a probability distribution implies a continuously defined range of possible values. Here, we have a limited number of values ("bins"), for each of which we're computing a number of occurrences. For this reason, we're using the term frequency in the established statistician's meaning, where it is the number of times a certain event has been observed.

The term *absolute frequency distribution* then refers to the total number of occurrences in each bin, the *relative frequency distribution* is the absolute frequency distribution divided by the total number of observations.

This usage is entirely in line with e.g. `https://en.wikipedia.org/wiki/Frequency_(statistics)`.

L822-826: This statement does not consistent with the figure 9(a), Fig.10(b,c) and Fig. 11(b,c). Please clarify this.

With the change of the ambient TI value, the damage distributions also have changed slightly. In general, high damage is not only induced in the main wind direction but also

from the high wake-induced turbulence. We hope to have clarified this in the explanation of the text.

(same answer as to comment 32. of reviewer 1)

L839-845, 891-895: Those lines are overlapping with other section, e.g. L401 to 405. Please consider to remove them.

Both lines passages are removed in course of the restructuring and the revision of results (See "Response to the structure and the readability").

Figure 14: Caption: "Annual progression over time for damage energy and profit for multiple optimized planning strategies", what do you mean by "damage energy". I think part of the sentence is misplaced. it should be :"Annual progression over time for damage and energy profit for multiple optimized planning strategies", right?

There was a comma missing between damage and energy. The corrected caption is now: *"Annual progression over time for accumulated damage, energy, and NPV for multiple optimized planning strategies"*. We hope this is now clear.

Section 5: some of the content can be moved to section 4.x where you show the results to make the text flow more smoothly and also increase the readability.

We revised the discussion section to a large extent. Mainly, we address the comments of reviewer 1 and from your side by moving discussions from previous sections to the discussion part. Nevertheless, the paragraphs which explicitly discuss the optimization results of the application example are moved to section 4 as you suggested. They are also shortened to some extent.